# Sphaeroptica: A tool for pseudo-3D visualization and 3D measurements on arthropods

**Aurore Mathys**[1,2,3☯*], **Yann Pollet**[1☯*], **Adrien Gressin**[4], **Xavier Muth**[4], **Jonathan Brecko**[1,2], **Wouter Dekoninck**[1], **Didier Vandenspiegel**[2], **Sébastien Jodogne**[5], **Patrick Semal**[1]

**1** Scientific Service of Heritage, Royal Belgian Institute of Natural Sciences, Brussels, Belgium, **2** Collections Management, Royal Museum for Central Africa, Tervuren, Belgium, **3** Documentation, Interpretation & VAlorisation of Heritage, ULiège, Liège, Belgium, **4** School of Engineering and Management Vaud, HES-SO University of Applied Sciences and Art Western, Yverdon-les-Bains, Switzerland, **5** Institute of Information and Communication, Technologies Electronics and Applied Mathematics (ICTEAM), UCLouvain, Louvain-la-Neuve, Belgium

☯ These authors contributed equally to this work.
* aurore.mathys@africamuseum.be (AM); ypollet@naturalsciences.be (YP)

**Data Availability Statement:** The code for the Sphaeroptica application is openly available on GitHub at the following URL: https://github.com/ypollet/Sphaeroptica-1. The application manual,

## Abstract

Natural history collections are invaluable reference collections. Digitizing these collections is a transformative process that improves the accessibility, preservation, and exploitation of specimens and associated data in the long term. Arthropods make up the majority of zoological collections. However, arthropods are small, have detailed color textures and share small, complex and shiny structures, which poses a challenge to conventional digitization methods. Sphaeroptica is a multi-images viewer that uses a sphere of oriented images. It allows the visualization of insects including their tiniest features, the positioning of landmarks, and the extraction of 3D coordinates for measuring linear distances or for use in geometric morphometrics analysis. The quantitative comparisons show that the measures obtained with Sphaeroptica are similar to the measurements derived from 3D μCT models with an average difference inferior to 1%, while featuring the high resolution of color stacked pictures with all details like setae, chaetae, scales, and other small and/or complex structures. Shaeroptica was developed for the digitization of small arthropods but it can be used with any sphere of aligned images resulting from the digitization of objects or specimens with complex surface and shining, black, or translucent texture which cannot easily be digitized using structured light scanner or Structure-from-Motion (SfM) photogrammetry.

## Introduction

Natural history collections serve as invaluable reference collections that play a crucial role in advancing scientific knowledge and understanding the natural world. Scientists use these specimens as a baseline for identifying and classifying new species, tracking changes in biodiversity, and studying the effects of environmental shifts. Additionally, natural history collections

along with example materials, can be accessed via dx.doi.org/10.17504/protocols.io.6qpvr352pvmk/v1 and the Sphaeroptica YouTube channel at https://www.youtube.com/@Sphaeroptica.

**Funding:** The development of Sphaeroptica, as well as current and future work, is funded by the DIGIT-4 program and the CANAHIST project of the Belgian Science Policy Office (BELSPO). This research is also supported by SwissCollNet from the Swiss Academy of Sciences (project no. SCN204-VD). The funders had no role in study design, data collection and analysis, decision to publish, or preparation of the manuscript.

**Competing interests:** The authors have declared that no competing interests exist.

offer a historical perspective, enabling researchers to trace the evolution and distribution of species and ecosystems. As we confront global challenges such as climate change and habitat loss, these reference collections become even more critical for informing conservation efforts and guiding sustainable practices [1].

The digitization of natural history collections is a transformative process that enhances accessibility, preservation, and utilization of biological specimens. As technology continues to advance, digitizing these collections becomes crucial in democratizing scientific knowledge and fostering global collaboration [2]. Digitization has also a positive impact on the economic value of the natural history collections [3]. Moreover, digitization contributes to the preservation of fragile specimens by reducing the need for physical handling. While 2D imaging was deemed sufficient for a long time, the trend is now shifting towards 3D digitization, as 3D enables virtual and more precise measurements of objects and specimens [4].

However, not all collections can be accurately digitized in 3D using contemporary available technologies. The digitization of insects, while offering numerous advantages, comes with its own set of limitations. One primary challenge is the inherent complexity and diversity of insect specimens, with variations in size, color, and structure which poses difficulties for the automated digitization processes. Achieving high-resolution and accurate digital representations of these intricate organisms remains a technical hurdle [5, 6].

Arthropods are not only small, but also feature antennas, setae, chaetae, scales, spines, small legs, thin wings—elements that can pose significant challenges for traditional 3D digitization techniques. These structures can be critical for taxonomic identification. Additionally, color texture can play a significant role in the taxonomic study of entomological specimens.

Offering the possibility of making 3D linear measurements on arthropods can play a pivotal role in advancing the taxonomy of arthropods, offering a powerful and quantitative approach to study morphometric variations within taxa. So far measurements in entomology have primarily relied on 2D data [7–10]. However, utilizing 2D projections for 3D-structured body parts can result in potential distortions and a significant loss of morphological information. This is mainly due to the inherent limitations of 2D measurements, which can induce deformations or biases due to the positioning of the specimen [10] or issues related to perspective and radial distortion [11]. This inadequacy stresses the importance of adopting advanced 3D measurement techniques for a more accurate and comprehensive capture of morphological details.

Several studies [5, 12, 13] have been dedicated to developing methods for digitizing entomological collections in 3D, with varying degrees of success. While it has been possible to digitize legs and antennas, setae and scales are seldom rendered, except perhaps through micro-CT (μCT) or nano-CT. However, these techniques can be expensive and inaccessible to many research teams. Moreover, they do not capture color and the metallic pins used to secure most insects in collection boxes can interfere with produced scan data [14].

Among the previous work made on the (pseudo) 3D digitization of arthropods, the ZooSphere [15], the Disc3D [12], and the scAnt [13] setups can be mentioned.

The ZooSphere project, developed between 2013 and 2015 at the Museum fur Naturkunde in Berlin (MfN), was probably the first system enabling the recording and viewing of entomological specimens from all angles with a high level of details [15]. ZooSphere employs focus-stacking techniques to capture between 100 and 144 views, through which the user can virtually navigate in a multi-view environment. The use of focus stacking allows for extended depth of field (EDOF). ZooSphere allows only visualization, but no measurements, and the project seems to be discontinued by MfN nowadays.

In parallel to the ZooSphere, Nyugen *et al.* [5] propose to produce 3D models of insects using volume carving (or shape from silhouette). The downside of this method is that the

concavities cannot be reconstructed. The models obtained are relatively simple, as small details cannot be reconstructed, but offer a photorealistic texture.

In the footsteps of the Zoosphere, the Disc3D solution was developed in 2018 [12]. Its authors developed proprietary hardware to automate image acquisition. Like the ZooSphere, they use EDOF images to generate 3D models using structure from motion (SfM), a method that enables 3D reconstruction from photographs. They developed a specific Mathematica tool to make the EDOF images compatible with the conventional photogrammetric pipeline. While their 3D models effectively captured details such as wings and some silks, the experimental method of Visual Consistent Mesh Generation parameter they used to obtain good results has been removed from the last versions of the used SfM software (Agisoft Metashape).

The scAnt project is very similar to the Disc3D, except that its authors have developed an open-hardware setup and have proposed open-source software [13]. As for the precedent studies using 3D models, they mention that small features and transparent structures are difficult to reconstruct in SfM. Wings are modeled from the pictures and are added to the 3D SfM model afterwards. The authors also indicate that the reflective surfaces can introduce noise in the models.

In this paper, Sphaeroptica 1.0 is introduced as a different workflow for the digitization of arthropods. Sphaeroptica offers a viewer for the sphere of pictures, similarly to ZooSphere or Quicktime VR, but leverages photogrammetry principles to place landmarks on 2D images and to perform 3D linear measurements. The advantage of this method is that it captures and preserves all the native details of the specimen in high-resolution images.

Sphaeroptica enables the measurements of details that cannot be easily rendered on 3D models, while avoiding the risk of parallax errors in the measurement performed on 2D images. Sphaeroptica is an open-source software designed to visualize a specimen in multiple orientations, to produce 3D landmarks that can be used to perform linear measurements and that can be exported in geometric morphometrics packages.

This paper also compares Sphaeroptica 1.0 to the traditional 3D approaches, establishing the accuracy and the advantages of this workflow, while highlighting some limitations.

## Method

### General approach

Sphaeroptica is an open-source software developed by Y. Pollet for his Master thesis at the Université catholique de Louvain [16] under the supervision of S. Jodogne, in the framework of a collaboration with the Royal Belgian Institute of Natural Sciences (RBINS). The goal of Sphaeroptica is to visualize a specimen under multiple angles, featuring the positioning of landmarks on the original stacked images instead of on a shape computed model, while taking in account the parallax deformation and allowing 3D linear measurements. This enables researchers to visualize and measure all visible details on the pictures and not only the one rendered in the approximated 3D model. Landmarks must be placed in at least two pictures for the triangulation process. The user can navigate through the sphere of pictures and choose the picture that is best suited for the placement of the required landmark. It is consequently possible to measure the distance between two landmarks that are not visible in the same image.

The images used with Sphaeroptica in this paper were acquired using the scAnt hardware developed by Plum & Labonte [13], but it is also compatible with images produced by other setups, such as the Disc3D setup [12], the Stackshot 3X combined with a DSLR [4], or a mirrorless camera [5].

Sphaeroptica is an alternate path to the SfM workflow. Sphaeroptica uses image matching and bundle adjustment algorithms to compute the 3D orientation pictures in space. However,

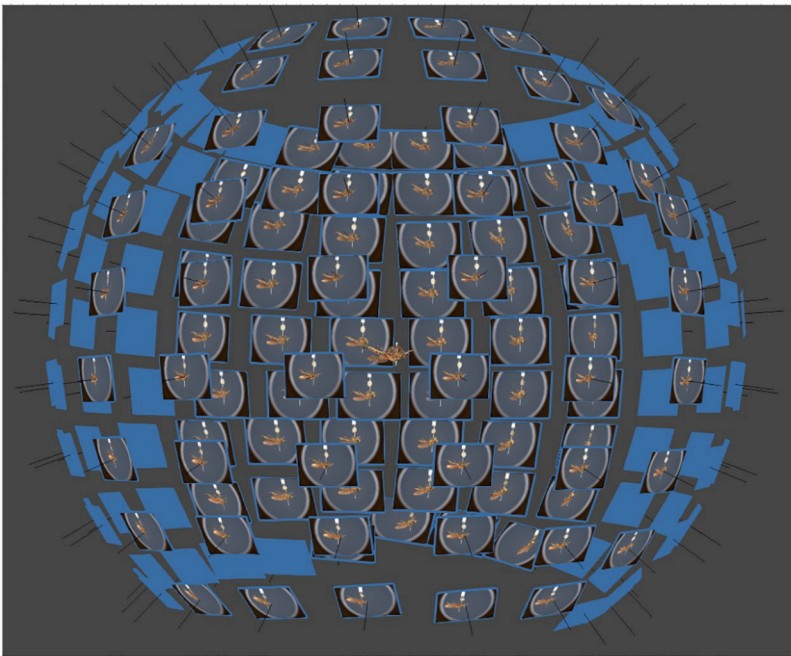

**Fig 1. Representation of the oriented of the oriented pictures in Agisoft Metashape Pro 2.0.2.**

it does not rely on the dense cloud, meshing, and texturing processes. Thanks to the camera calibration and to the accurate 3D orientation in space of the pictures using the SfM algorithm, Sphaeroptica is able to compensate for the parallax measurements error. This produces a sphere of oriented high-resolution 2D images (Fig 1), with the possibility to place 3D landmarks and to compute distances between landmarks of interest.

Fig 2 summarizes the workflows of SfM and Sphaeroptica. This figure shows that both techniques share a common part, but provide two different pathways.

In this research we will compare the results of Sphaeroptica to other standard 3D digitization methods such as automatic photogrammetric process Structure from Motion—Multi View Stereo (SfM-MVS), structured light scanning (SL), and micro computed tomography (μCT).

To this end, a comparative analysis was conducted between the 3 most commonly used methods, i.e., photogrammetry, structured light, and μCT.

The photogrammetry method is SfM-MVS. SfM-MVS can recreate 3D objects from photographs taken at different positions. It is a robust and well-established non-contact method for documenting specimens in 3D. The advantage of SfM-MVS over classic photogrammetry approaches is that it allows the computation of a 3D model from a set of overlapping, unoriented, and uncalibrated images of unknown positions [17–19]. SfM-MVS photogrammetry is composed of 3 main steps: Tie points are first detected using image matching algorithms such as SIFT or SURF, then the bundle adjustment phase called SfM is applied, and finally the MVS algorithm generates a dense cloud [18]. The next steps consist in meshing and texturing, which are not specific to photogrammetry, but are usually implied when talking about photogrammetry models, as it is the case here.

As shown upon reviewing the literature on 3D digitization of entomological collections, photogrammetry is often chosen, partly due to its cost-effectiveness and ease of application [4, 5, 12, 13].

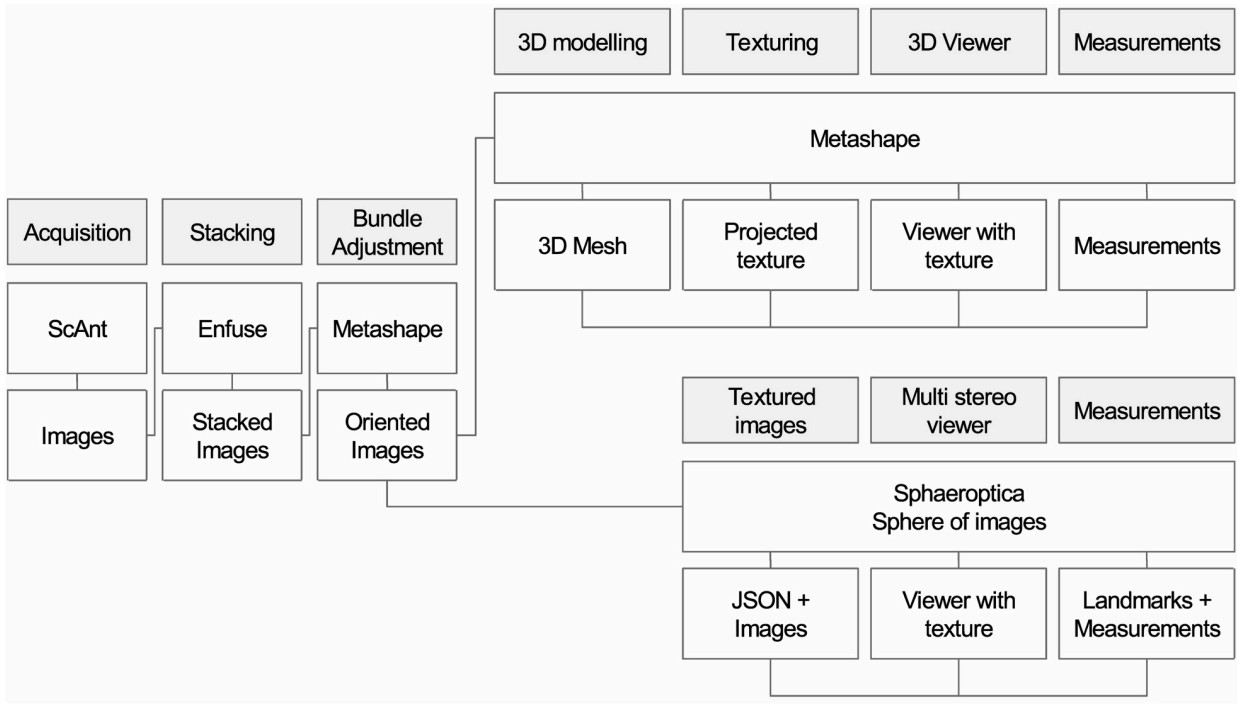

**Fig 2. Workflow of Sphaeroptica compared to classical SfM workflow.**

The image matching and SfM algorithm employed in photogrammetry are similar to those used by Sphaeroptica. However, for photogrammetry, the next step applied is the cloud densification or the mesh generation, while these are not in use within Sphaeroptica.

This study uses the Agisoft Metashape Pro package, one of the most popular proprietary software to digitize collections using photogrammetry. Open-source alternatives such as Alice-Vision or COLMAP could also be considered to compute the positioning and orientation of the stacked pictures, and to export the pose estimation.

Structured light (SL) consists in the projection of a light pattern onto the specimens. The deformation of the light pattern is captured by one or more cameras, which allows the calculation of the 3D surface by triangulation.

Some SL scanners are precise enough to capture very large entomological specimens, but this is seldomly used. We were not able to find literature about it, with the exception of an experimental digitization in Brecko & Mathys [4]. This can be explained by several reasons. Firstly, there are fewer SL scanners adapted to small-sized specimens that deliver enough precision. Moreover, most of the scanners capable of capturing small specimens lack the ability to acquire color texture. Secondly, insects are often reflective and/or dark, which are two of the material properties that pose issues for SL digitization. Finally, the transparency of the wing is also a challenge. Manufacturers will often propose to spray the specimens with a mattifying coating to obtain a good model. It is indeed a possibility, but this could represent a risk for the specimens. In our experience, we definitely recommend avoiding this process for valuable specimens. We chose to include SL in our study because metric results from SL are supposed to be reliable. We used the Artec Micro SL scanner, which is one of the few scanners with the required level of accuracy and the ability to capture color information. Moreover, this scanner can digitize a complete specimen automatically, only requiring a single switch of the position

of the specimen and performing an automated alignment of the two sides. Its theoretical precision is up to 10 μm.

Finally, micro-Computed Tomography (μCT) was also considered in this study. μCT is a non-destructive imaging technique using X-rays allowing the tridimensional digitization of both the external and internal structures of a specimen. μCT can be applied to entomology. Nevertheless, for insects pinned on a metallic needle, the method has its limitations. Indeed, due to the large difference in density between the metal and the body composition, "metal artifacts" appear on the digitized specimens [14]. As a consequence, the portion of the specimens that is located around the pin cannot be reconstructed in 3D. Another limitation of μCT is that it cannot capture the color texture of the specimen. Ijiri *et al.* [20] proposed to reproject texture from photographs onto μCT models to overcome the lack of color. Nonetheless, μCT scanners are expensive and not widely available, their use requires specific technical knowledge, and the μCT segmentation can be time-consuming. Not all μCT setups are suitable for entomological collections and, often, researchers have to use synchrotron μCT facilities which are even less accessible [12]. Accurate segmentation of μCT data of arthropods can require very advanced knowledge as some structures such as wings are extremely flat and weakly contrasted under X-ray [21]. Furthermore, μCT can be invasive as it sometimes requires staining with contrast agents to obtain good results [22].

In this study, the conventional cone-beam μCT EasyTom 150 from RX solutions (Chavanod, France) was used, without staining the specimens as we are looking for digitization workflows which can also be applied on type specimens.

In this research, four arthropods were digitized using all four techniques to evaluate the quality of their respective digital twins, using both visual evaluation (details visible on the model, rendering artifacts) and quantitative landmarks positioning and measuring. μCT reconstructions were used as a metric reference, even if the accuracy of the reconstructed 3D models can also be affected by the threshold values used during the segmentation of such images.

The selected specimens of arthropods present different challenges (Fig 3):

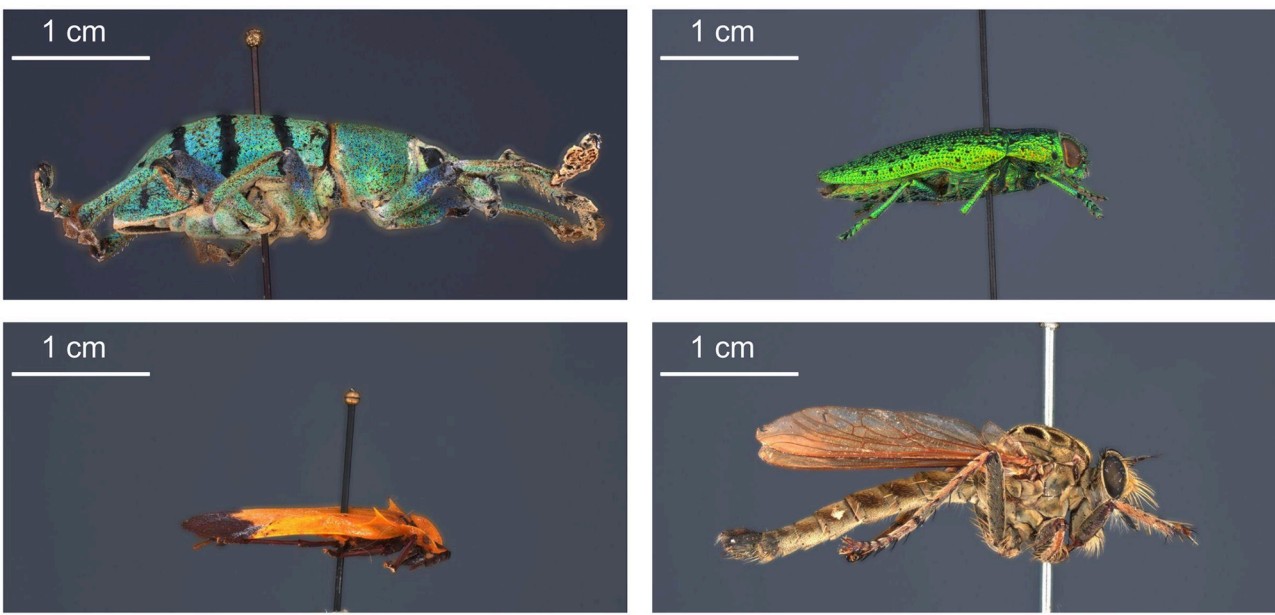

**Fig 3. Case studies:** *Eupholus schoenherri, Ovalisia sp., Lycus rostratus* and *Philodicus* sp.

1. A relatively mat and coloured Coleoptera: *Eupholus schoenherri* (Coleoptera: Curculionidae). This specimen does not present too many difficulties for SL. The body of this arthropod is approximately 27 mm in length. It is relatively simple to digitize as it is not very reflective or iridescent and does not present many setæ or transparent structures. The μCT was performed using 25W, 90kV, 275uA, with a voxel size of 29.05μm for 1440 projections.

2. A coloured and reflective Coleoptera: *Ovalisia* sp. (Coleoptera: Buprestidae). This specimen of 18 mm is difficult to digitize using SL and SfM due to the iridescent and highly reflective materials. The μCT was performed using 9.4W, 75kV, and 125uA, with a voxel size of 14.21μm for 1440 projections.

3. A flat beetle but not reflective and without many setæ: *Lycus rostratus* (Coleoptera: Lycidae). The challenge here is the flatness: Flat surfaces can often be difficult to reconstruct in a 3D mesh. It is also a challenge for the 3D orientation of the pictures corresponding to the upper and lower sides of the specimen. The specimen is approximately 16 mm in length. The μCT was performed using 10W, 60kV, 166uA, with a voxel size of 14.20μm for 1440 projections.

4. A robber fly with many setæ but also translucid and iridescent wings and iridescent eyes: *Philodicus Loew* sp. (Diptera: Asilidae). This represents the ultimate challenge for SL and SfM. It is even difficult to reconstruct a complete 3D mesh from a μCT scan. The specimen has a length of approximately 25 mm. The μCT was performed using 4W, 40kV, 100uA, with a voxel size of 21.22μm for 1440 projections.

## Sphaeroptica workflow

The images were acquired with the scAnt hardware (Fig 4). To know more about that part of the process, please refer to the paper of Plum and Labonte [13]. Prior to the acquisition, the labels beneath the specimens are removed and a 7 mm hexagonal metallic bead is placed on the needle below the specimen to act as calibration object providing the scale.

The pictures were taken by the industrial 20Mpx FLIR camera with a 1 inch sensor combined to a 35mm HD lens, were stacked using the Enfuse open-source package that is part of the scAnt workflow, and were imported into Agisoft Metashape Pro 2.0.2. The theoretical pose of the images has been estimated according to the position of the specimen relative to the camera. Thus, the a priori pose of the cameras are known, but this pose does not take into account any derivation due to the friction or to the mechanical imperfections in the assembly. Additionally, as we use focus stacking, the distance to the gimbal rotation centre of the resulting stacked image is not known and has to be estimated. Therefore, to attain more accurate pose estimations, we refine the placement of the camera using the SfM orientation process. Prior to orientation, the distance between the two reference points on the hexagonal scale (Fig 5) was measured (6.69mm) using a calliper and a Keyence Wide-Area 3D Measurement System VR-5000. Then we launched the bundle adjustment of the pictures. The hexagonal shape is also used for flat specimens that would be difficult to orientate otherwise due to the small number of key points that can be identified.

The intrinsic and extrinsic camera parameters were computed and exported from the photogrammetry software (Agisoft Metashape Pro 2.0.2). The intrinsic parameters correspond to the camera characteristic and the extrinsic parameters define the position of the camera in space [23]. The result of the bundle adjustment generates accurate camera positions and scales with the intrinsic and extrinsic parameters (Fig 6). These data are saved and imported in Sphaeroptica together with the pictures.

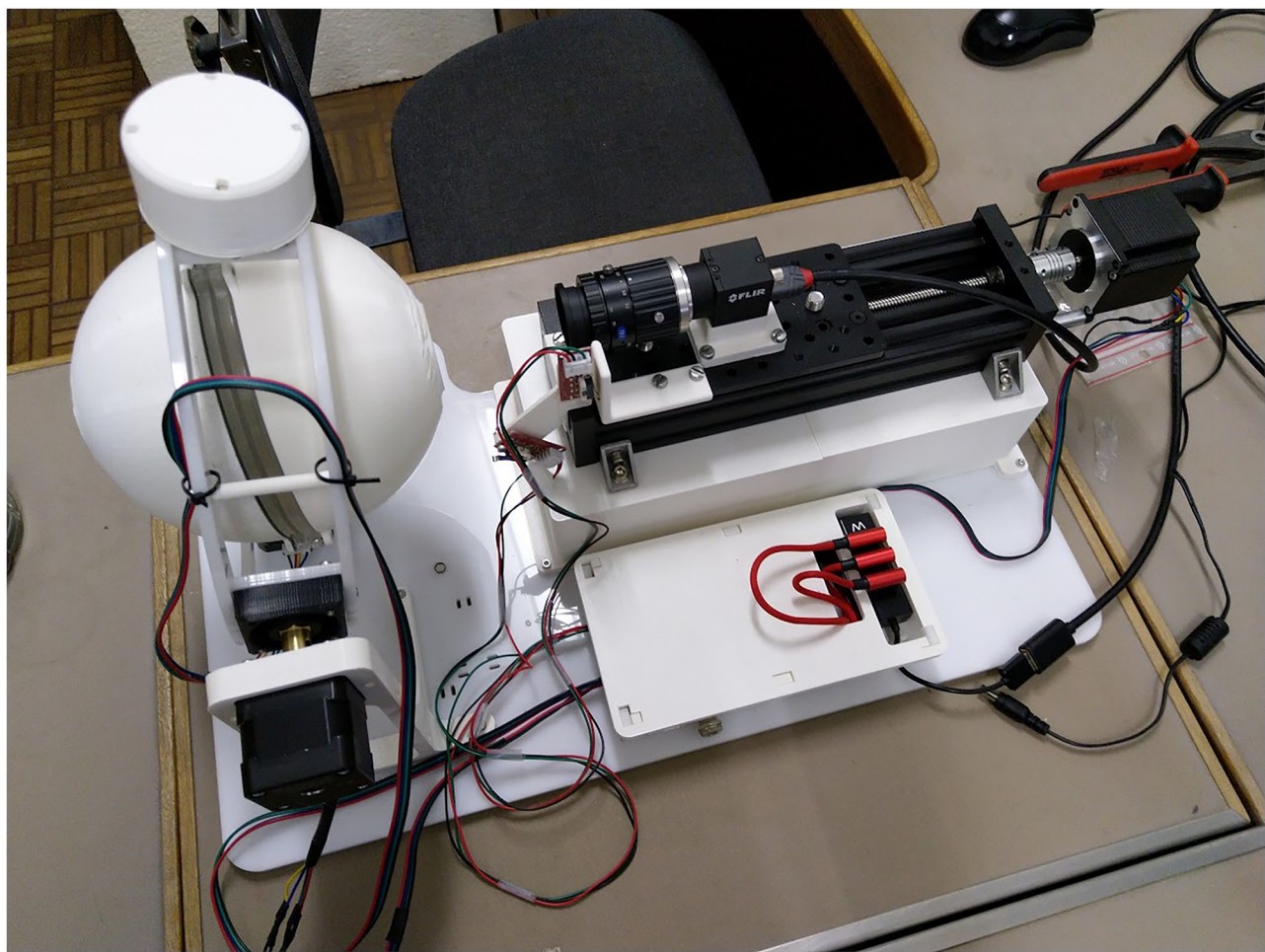

**Fig 4. Picture of the RBINS scAnt setup.** Based on the paper of Plum and Labonte [13].

The two files, containing the extrinsic and intrinsic parameters, allow the positioning of the cameras in 3D space to compose the sphere of oriented pictures.

The first user interface of Sphaeroptica provides a pseudo 3D viewer to navigate the sphere of oriented pictures (Fig 7). Given a 3D position over the sphere, a nearest-neighbor search is used to retrieve the reference image and to display it to the user. The user can therefore define registered pictures corresponding to the frontal, posterior, superior, inferior, left lateral, and right lateral views of the specimens, allowing fast navigation between the different standard views. These views are used by scientists for the publications of new taxa and/or to identify unknown specimens.

A second user interface can be activated by selecting a specific view in the sphere of pictures (Fig 8). In this case, Sphaeroptica opens a new window with the full resolution image. The user can then place landmarks at precise locations in the image. Landmarks must be placed in at least two pictures for the triangulation process. The list of the landmarks, together with their label and referenced color, is displayed on the right menu of the application. If landmarks are identified on at least two pictures, measurements between two landmarks can be computed by triangulation. The distance between two landmarks is defined up to a scale factor that can be

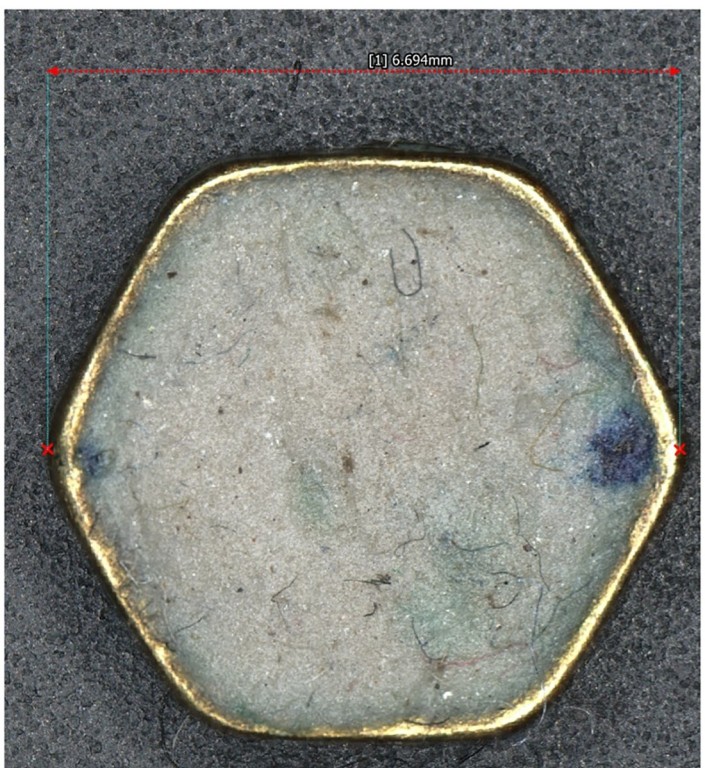

**Fig 5. Hexagonal metallic bead used as a scale.** The bead has been filled with plasticine to stay fixed on the needle.

retrieved by measuring the size of a calibration token in the image (in our case, the hexagonal metallic bead).

The positions of the landmarks are computed using the triangulation algorithm that was defined by Hartley and Sturm [24]. The principle is that the position of landmarks on each

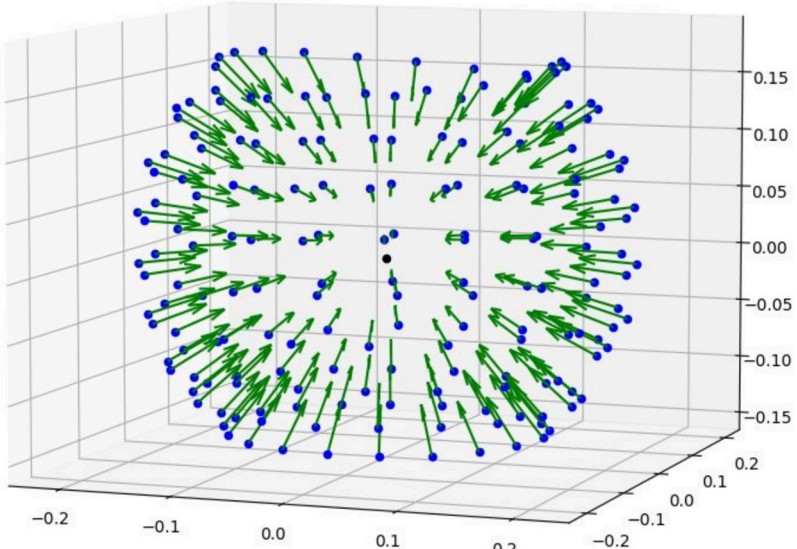

**Fig 6. Vector matrix of oriented pictures in Sphaeroptica.**

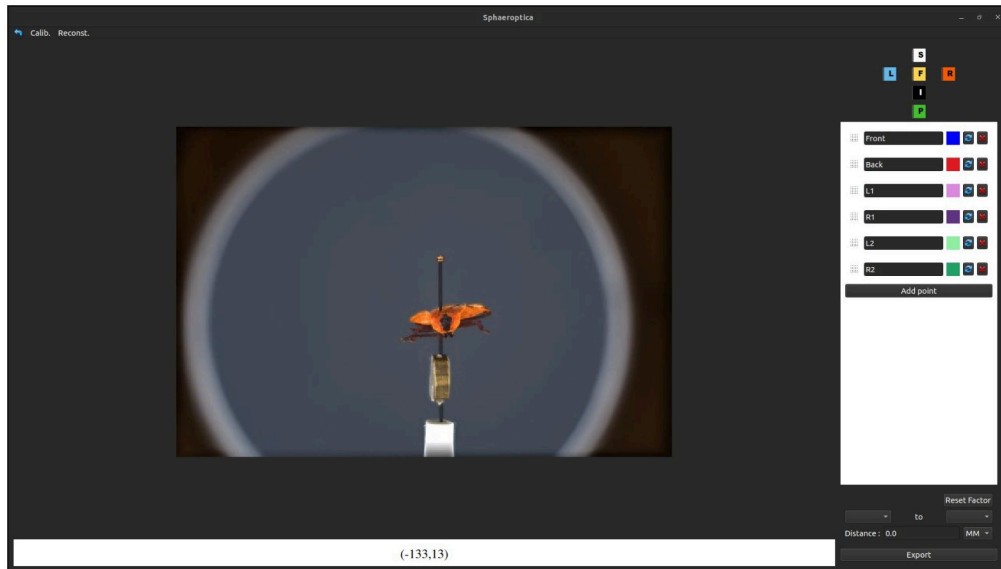

**Fig 7. Main interface of Sphaeroptica.** It offers a pseudo 3D viewer to navigate the pictures that were taken over the sphere of image acquisitions around the specimen.

view gives us a ray that goes through the universe crossing the camera and the image plane at that pixel position (Fig 9). If at least two rays are available, the intersection of the rays corresponds to the 3D coordinates of the landmark. Unfortunately, because of noise in the images and human error in the input, the rays will generally not intersect exactly at a precise point. To

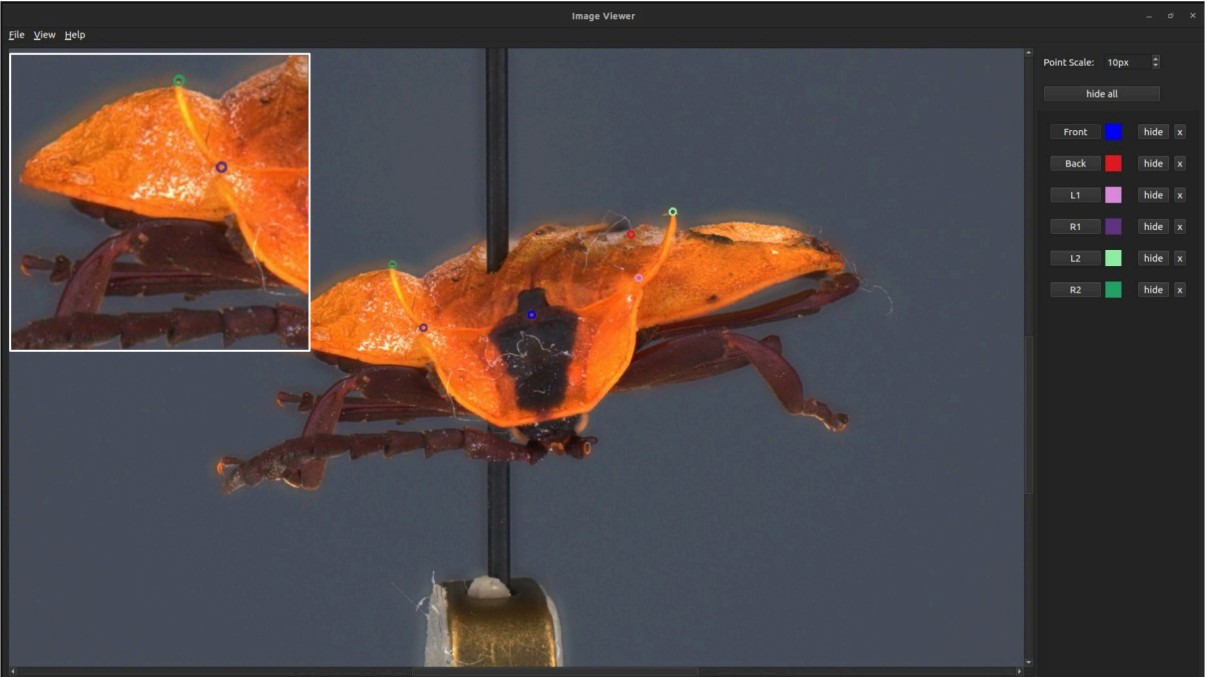

**Fig 8. Sphaeroptica displaying a high resolution picture, with the display of the landmarks.** A detailed view with a zoom of the picture is also available.

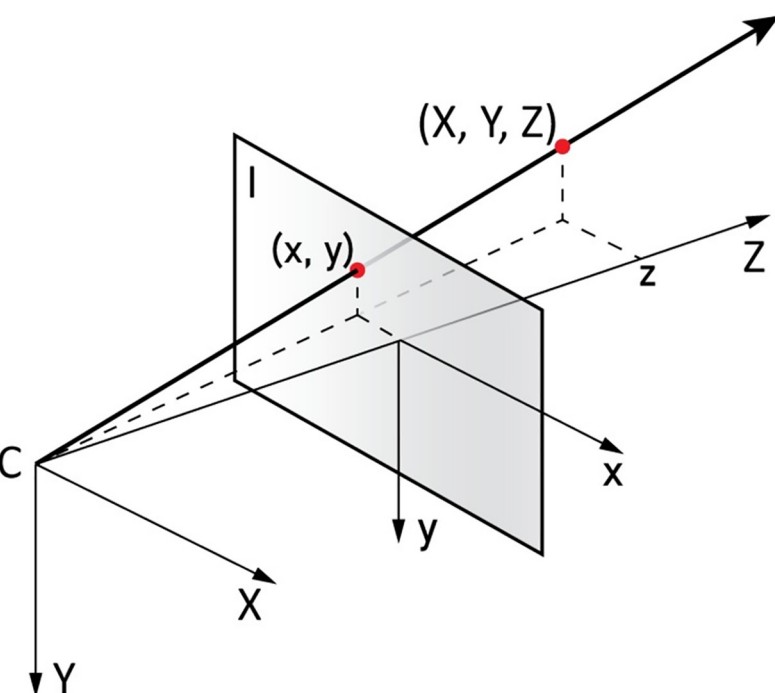

**Fig 9. Back projection of a pixel.** C is the camera center and I the image plane (after Solà, 2017 [25]).

solve that problem, a least-squares optimisation algorithm is used to find the 3D point that is the closest to all the rays (Fig 10).

The data related to the landmarks (i.e., the labels, the (X,Y,Z) values, and the rescaled (X,Y, Z) values after internal calibration) can be exported as a CSV file that can be reused in any external application such as a spreadsheet application, a 3D software or a Morphometric Geometric software (Fig 11).

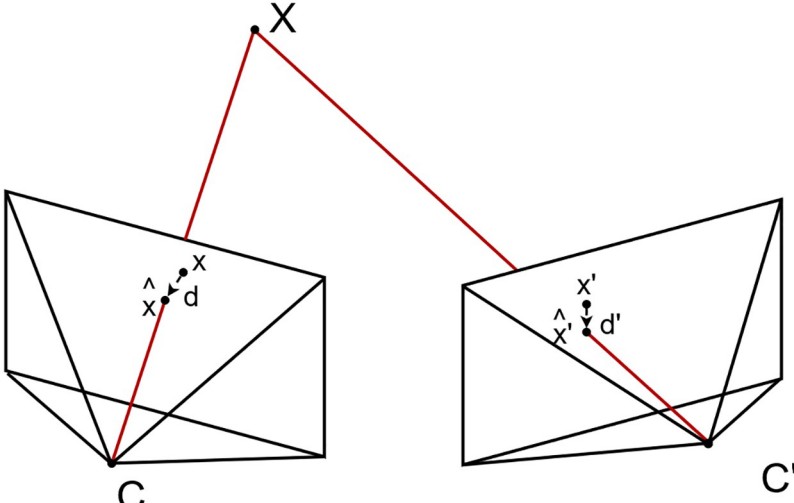

**Fig 10. Triangulation (after Hartley & Zisserman, 2004 [26]).**

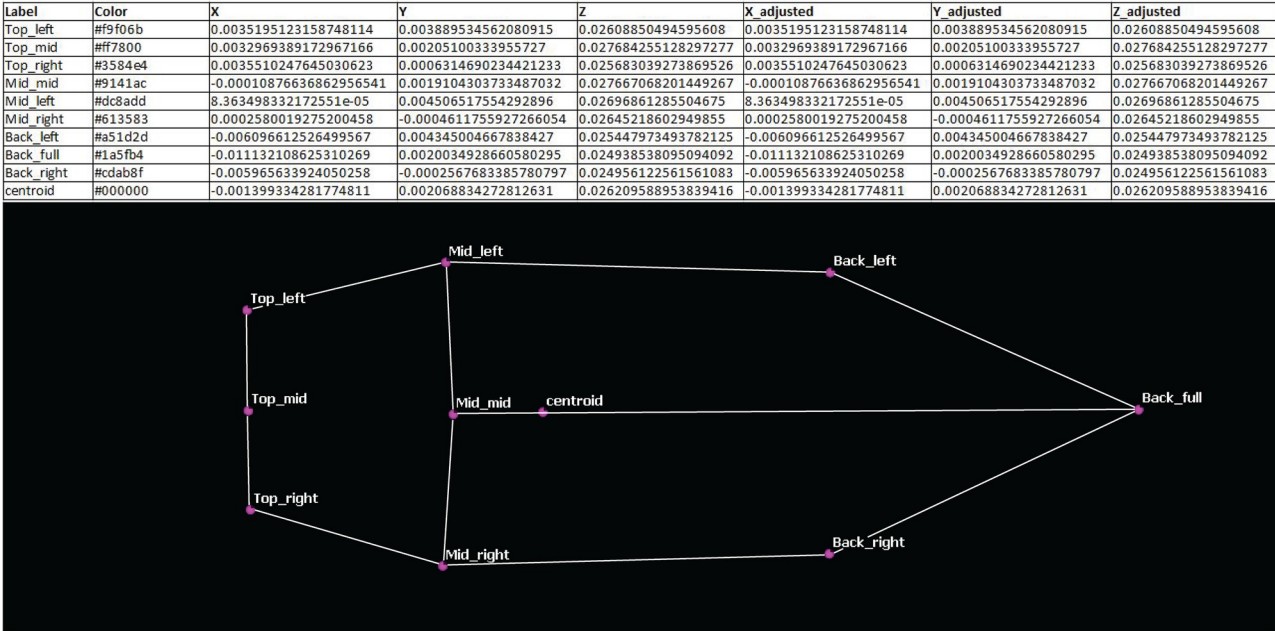

**Fig 11. Landmarks data exported as CSV and orthogonal projection of the shape derived from the 3D landmarks using CloudCompare.**

In the following sections, we measure specimens digitized using the different techniques and we compare them to the measurements computed by Sphaeroptica. As the insects were moved multiple times to the various digitization setups, our measurements focus on the main body, excluding the extremities of legs, to avoid any potential shift between captures from the different setups or during the digitization process.

## Results

The 4 specimens were digitized in 3D using 3 different techniques (μCT, SL, and SfM) to create reference data. A comparison of the results obtained by the different techniques was achieved using both a qualitative (and subjective) evaluation and a quantitative analysis of the 3D data and associated measures.

### Qualitative comparison

**Assessing visualizations for color and texture accuracy.** Each 3D imaging technique presents unique capabilities and limitations: μCT, SL, photogrammetry, and Sphaeroptica will yield distinct outcomes. μCT is able to produce a 3D mesh but does not capture color information. SL and photogrammetry both produce 3D meshes with color texture information. Meanwhile, Sphaeroptica specializes in image visualization without a 3D mesh, yet remains able to provide precise 3D reprojections (Fig 8).

Among the three methods that enable color information, the quality and level of details differ. Specifically, the SL model offers a texture with poor contrast and few details. Photogrammetry provides a more contrasted texture with additional details and more accurate color. Finally, the image visualization of Sphaeroptica allows for the display of all details, very sharply and with accurate color representation (Fig 12).

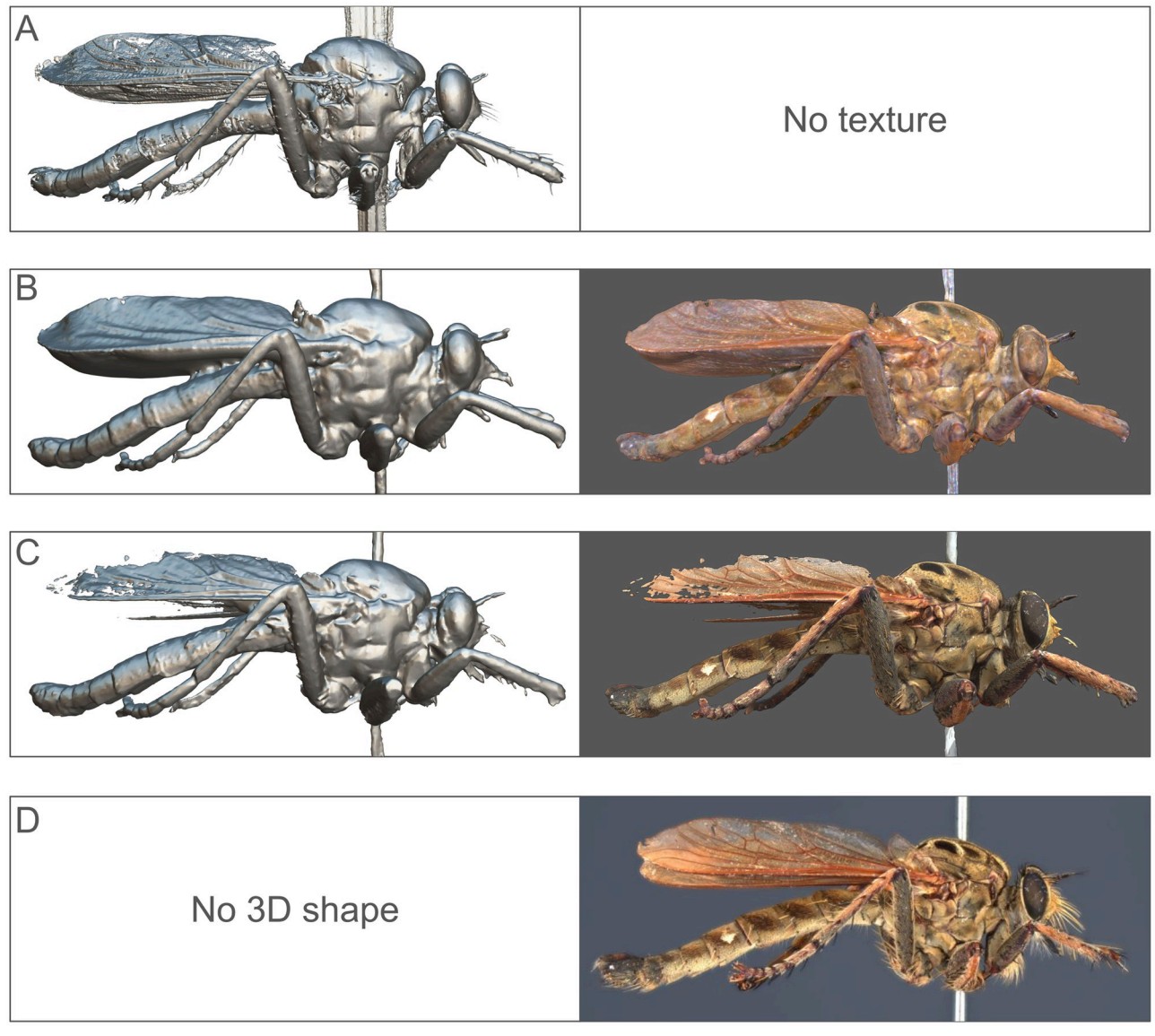

**Fig 12. Comparison of the shape and texture.** Comparison of shape and texture produced by (A) μCT (shape only), (B) Artec micro SL, (C) Photogrammetry SfM, and (D) Sphaeroptica (texture only).

**Qualitative comparison between the 3D surface models.** We conducted a qualitative visual comparison of the 3D models obtained through the three different techniques to assess their differences and choose which 3D model will be used as a metric reference in the quantitative comparison.

*Eupholus schoenherri*. Upon conducting a visual analysis of the Eupholus, the 3 models are relatively similar, but the μCT model exhibits finer details, especially for the antennas (Fig 13).

*Ovalisia* sp. and *Lycus rostratus*. For the *Lycus rostratus* (Fig 14) and the *Ovalisia* sp. (Fig 15), the SL model was not able to capture all the limbs. SfM and μCT are able to capture the full legs and antennas, but the μCT model exhibits more details and sharper features.

The SL model of the *Lycus rostratus* is very smooth and the wings are incomplete (Fig 14).

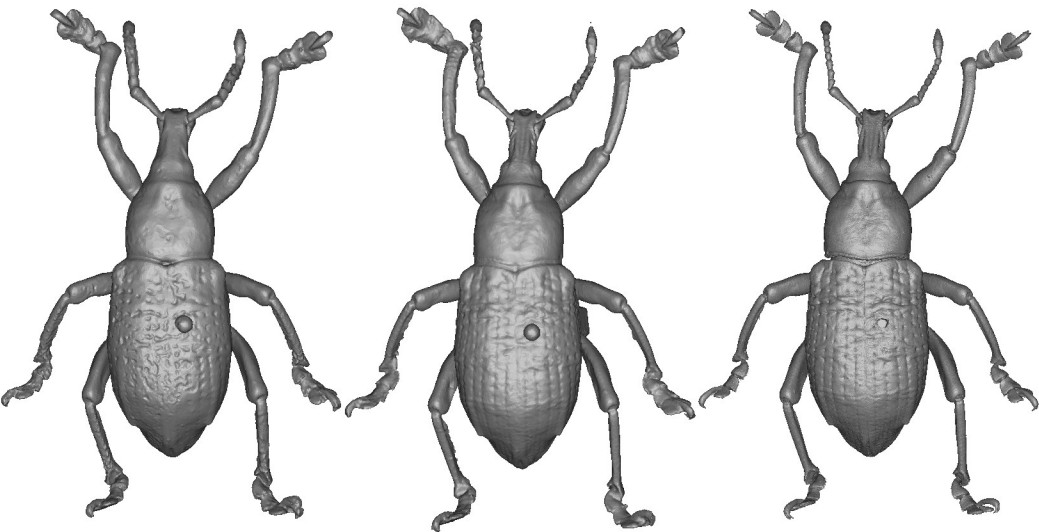

**Fig 13. 3D model of *Eupholus schoenherri*.** Left: SL. Center: SfM. Right: μCT.

*Philodicus* sp. In the case of the robber fly, the μCT model is clearly superior in terms of rendered details. It is possible to observe setæ and sharper structures, which contrasts with the other techniques. Nevertheless, the wing cells are partially incomplete, even if most of the veins are present. SfM and SL have neither been able to render the wings properly and their models are generally more smoothed (Fig 12).

Thus, upon conducting a visual analysis comparing μCT, SfM, and SL models, we can observe some trends: The SL models exhibit the lowest level of detail. In contrast, the μCT models emerge as the most intricate and precise. The SfM models are more complete than the

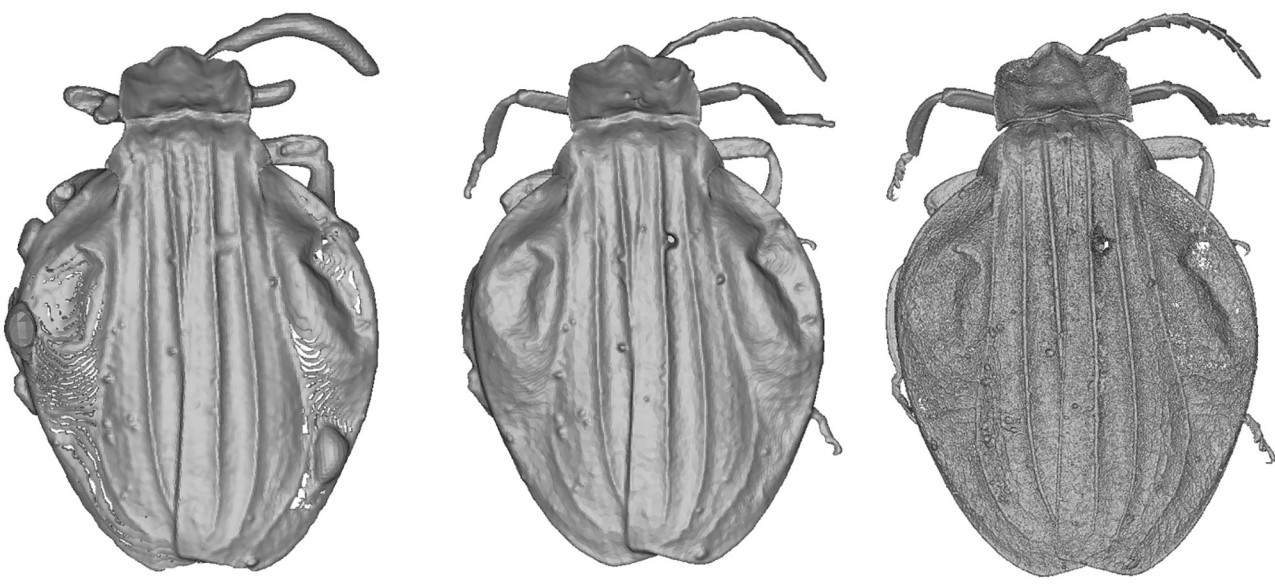

**Fig 14. 3D model of *Lycus rostratus*.** Left: SL. Center: SfM. Right: μCT.

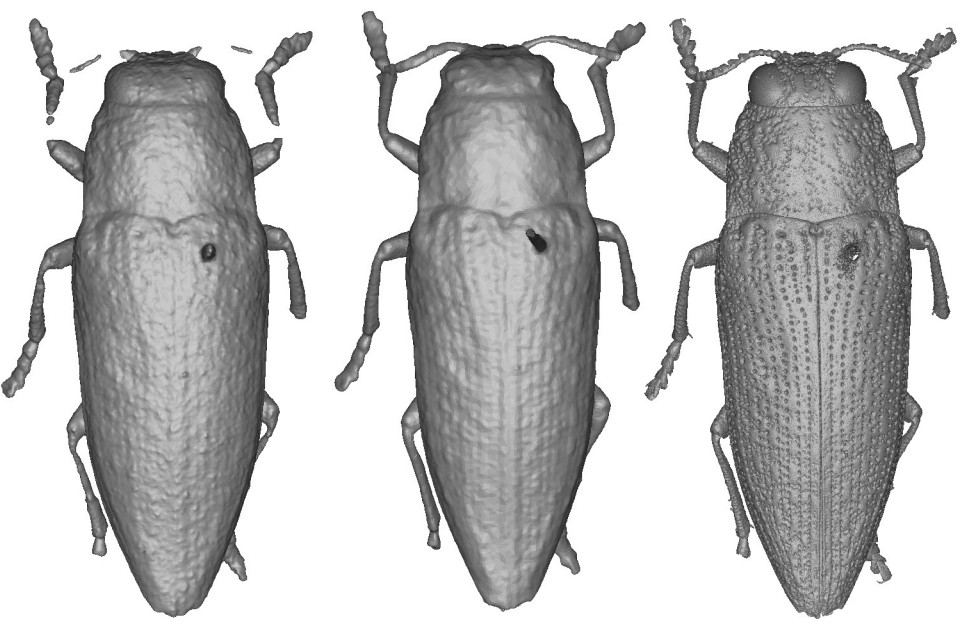

**Fig 15. 3D model of *Ovalisia* sp.** Left: SL. Center: SfM. Right: μCT.

SL models, but are much less detailed than the μCT models. We can conclude that the most accurate reference for metric comparison of landmarks are the μCT models.

As can be observed, it is difficult to obtain an accurate model of the wings or setæ due to their low density, even with μCT. It is possible that other μCT setups could generate better results, but the μCT modality evaluated in this study (RX Easy Tom) was not able to produce better images. Nevertheless, the superior quality of the μCT 3D surface model makes it the one to use as the gold standard to which Sphaeroptica measurement accuracy and precision will now be compared.

**Overview of 3D imaging methodologies and color representation.**   As explained in the previous section, the μCT scans produced the most detailed surface 3D models. Nevertheless, some very thin structures of the wings or some setæ are not always rendered correctly. We can also observe the 3D rendering of the needle is affected by metal artifacts and produces an inaccurate reconstruction of this part of the specimen. Moreover, the μCT does not produce color texture information.

The SL models produced the less detailed surface models. The fusion algorithms of the Artec Studio created a model without any gap and with smoothed details, which is an advantage to prepare 3D models for educational purposes, but not for scientific study. The quality of the texture is good for a SL scanner and can be improved using an imaging software, but it is not as good as the photorealistic texture obtained with the photographic approach.

The photogrammetric 3D models produced 3D surface models with a higher level of detail than SL, but still very smoothed in comparison to the μCT. Some of the small and thin structures are skipped during the meshing process. The texture is very good but is missing when the 3D shape is not correctly rendered.

The Sphaeroptica workflow does not produce any 3D model but the quality of the pictures allows one to see the small structures of the specimen like setæ or wings details (Fig 16).

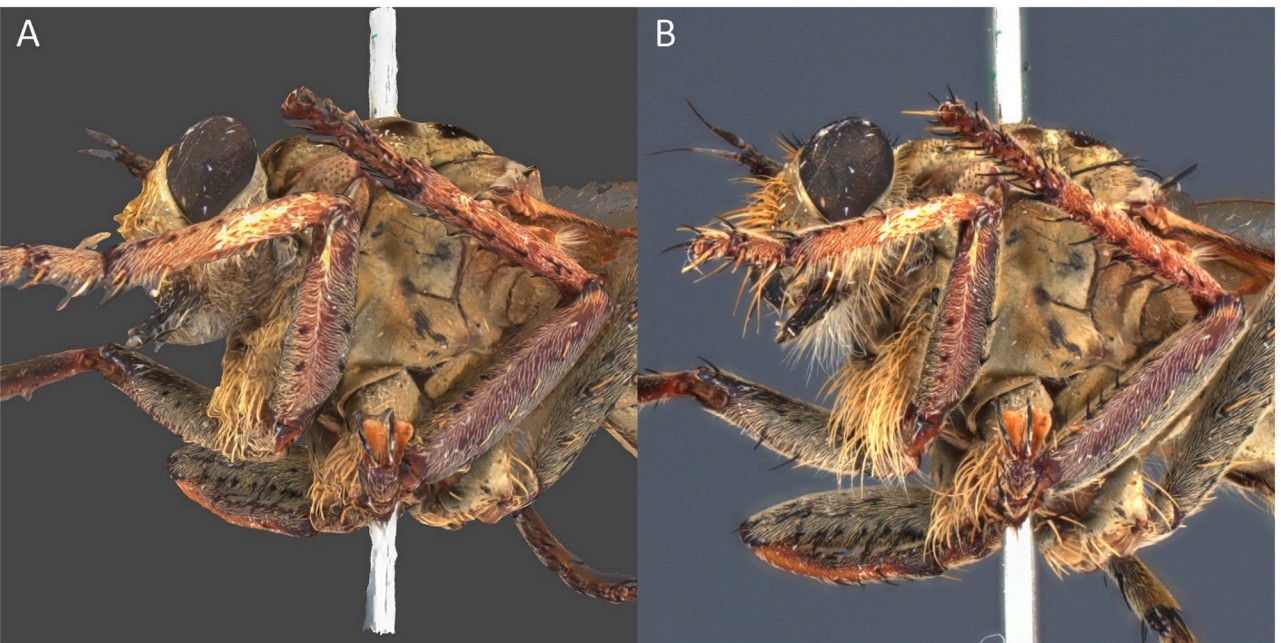

**Fig 16. Detail view of *Philodicus sp*. (A) In SfM, (B) In Sphaeroptica.** The SfM model cannot render as much detail as Sphaeroptica.

## Quantitative comparison of the techniques using landmarks

In order to review the metric accuracy of Sphaeroptica, it was decided to measure the distance between several landmarks on each model to estimate if the techniques allow for accurate measurements.

In theory, stacking can induce error in the measurements. As mentioned by Olkowicz *et al.* [27], the reconstruction of perspective during focus stacking is erroneous. It is what they called stacking distortion. However, their investigation led to the conclusion that the error introduced by stacking distortion in SfM is minimal. Consequently, they determined that FS-SfM remains a viable method for measurement purposes.

For each specimen, we tried to have landmarks in 3 different directions (X, Y, Z) that could be identified on all datasets. Most of the landmarks are placed on the main body of the insect, because legs could have moved between the different captures. Each distance between two landmarks was measured 6 times, 3 times by operator A (AM) and 3 times by operator B (YP), to take into account the intra- and inter-operator variability of measurement.

In 3D, landmarks can be directly positioned on the 3D models. In Sphaeroptica, landmarks must be positioned on a minimum of 2 pictures to be triangulated to the rest of the pictures and produce XYZ coordinates allowing to measure a distance between two 3D landmarks (Fig 17).

μCT was designated as our reference standard, considering it as the most precise 3D model.

The measurements were rounded to one hundredth of a millimeter, which corresponds to the precision of a digital caliper. The measurements are ordered in the tables from the smallest to the biggest values for each operator.

The average of these measurements are there compared to the average of the μCT standard using the following formula:

$$\mu\mathrm{CT}_{\mathrm{Diff}}(\%) = \frac{Avg(technique) - Avg(\mu CT)}{Avg(\mu CT)}, \qquad (1)$$

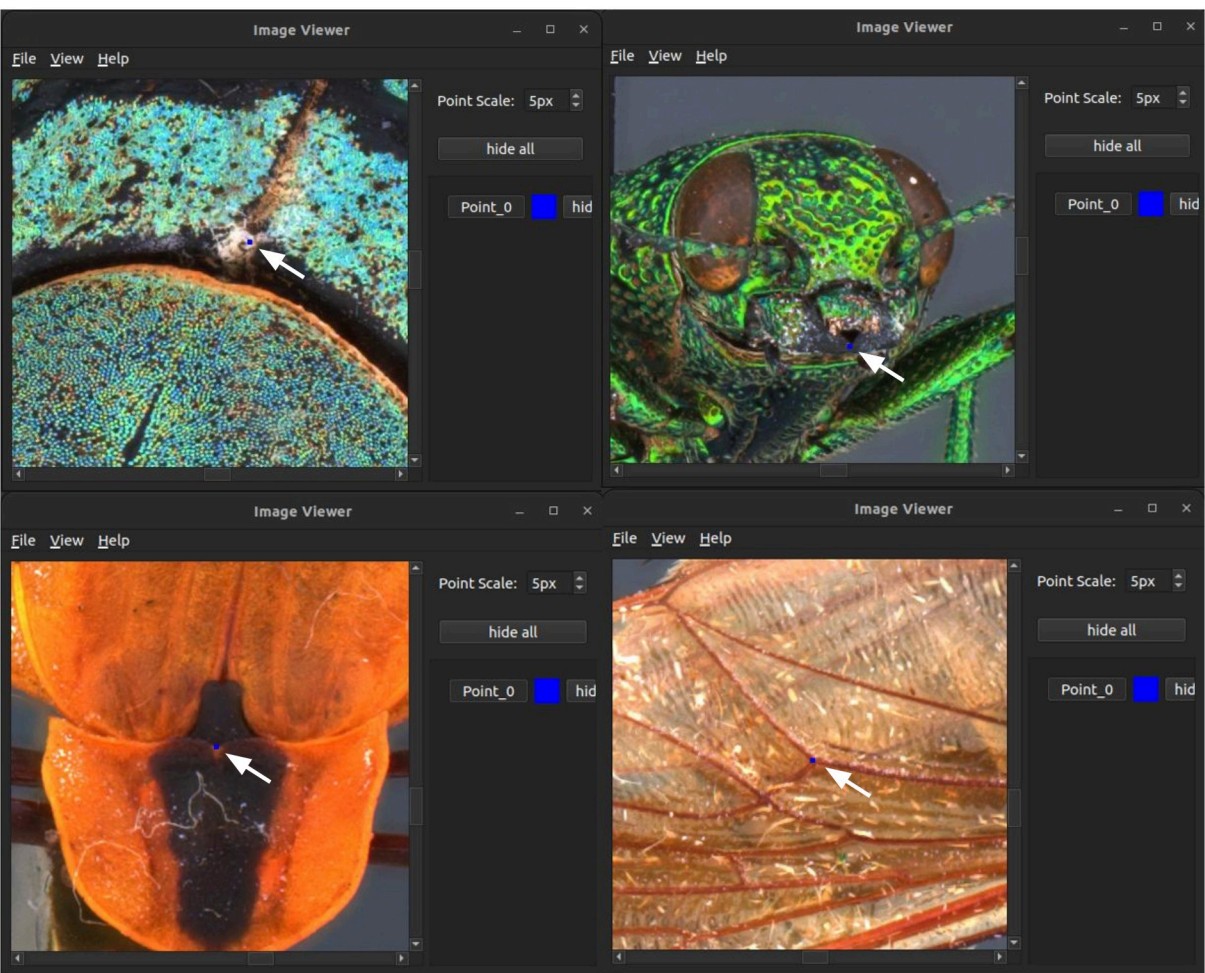

**Fig 17. Landmarks positioning in Sphaeroptica.**

The difference between operators is obtained using the same approach:

$$A - B_{Diff}(\%) = \frac{Avg(OperatorA) - Avg(OperatorB)}{Avg(OperatorB)}, \tag{2}$$

The dispersion of the measurements for each technique is estimated using the measurements of both operators using the following formula:

$$\frac{MAX - MIN}{AVG}(\%) = \frac{Maximum_{value} - Minimum_{value}}{Average}(technique) \tag{3}$$

***Eupholus schoenherri***. For Eupholus schoenherri, three distances were measured (Fig 18, Table 1). Table 2 shows that Sphaeroptica and SfM measurements are on average smaller than SL and µCT (around -2.3% to µCT), which suggests a possible error in scaling. The same range of difference is observed by the two operators, suggesting a real difference instead of an individual error measurement.

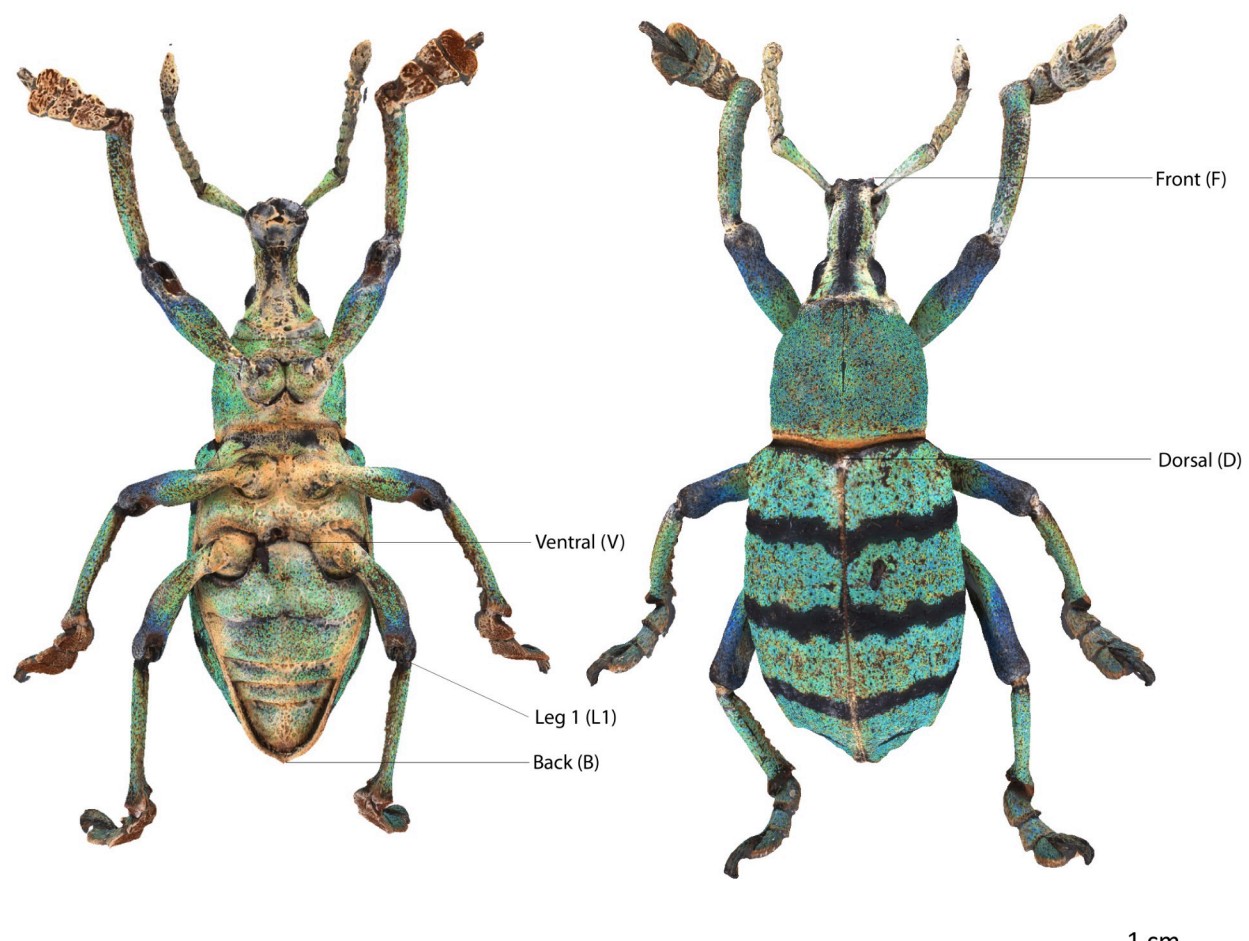

**Fig 18. Eupholus schoenherri.** Landmarks position on the Sfm model.

**Table 1. Landmark measurements (in mm) for *Eupholus schoenherri*.**

| | Operator A | | | | | Operator B | | | | | A-B | A and B |
|---|---|---|---|---|---|---|---|---|---|---|---|---|
| | **A1 min** | **A2** | **A3 max** | **Avg** | **Diff µCT** | **B1 min** | **B2** | **B3 max** | **Avg** | **Diff µCT** | **Diff** | **(MAX-MIN) / AVG** |
| **F-B** | | | | | | | | | | | | |
| *µCT* | 27.34 | 27.37 | 27.38 | 27.36 | – | 27.38 | 27.46 | 27.51 | 27.45 | – | -0.32% | 0.62% |
| *SL* | 27.45 | 27.50 | 27.53 | 27.49 | 0.48% | 27.48 | 27.50 | 27.52 | 27.50 | 0.18% | -0.02% | 0.29% |
| *SfM* | 26.73 | 26.78 | 26.79 | 26.77 | -2.18% | 26.71 | 26.74 | 26.76 | 26.74 | -2.60% | 0.11% | 0.30% |
| *Sphaeroptica* | 26.68 | 26.80 | 26.85 | 26.78 | -2.14% | 26.71 | 26.75 | 26.80 | 26.75 | -2.54% | 0.09% | 0.64% |
| **V-L1** | | | | | | | | | | | | |
| *µCT* | 9.48 | 9.56 | 9.63 | 9.56 | – | 9.55 | 9.56 | 9.59 | 9.57 | – | -0.10% | 1.57% |
| *SL* | 9.55 | 9.70 | 9.71 | 9.65 | 1.01% | 9.41 | 9.49 | 9.58 | 9.49 | -0.77% | 1.69% | 3.13% |
| *SfM* | 9.20 | 9.21 | 9.25 | 9.22 | -3.52% | 9.31 | 9.34 | 9.44 | 9.36 | -2.13% | -1.53% | 2.58% |
| *Sphaeroptica* | 9.10 | 9.20 | 9.30 | 9.20 | -3.73% | 9.06 | 9.26 | 9.36 | 9.23 | -3.55% | -0.29% | 3.26% |
| **D-V** | | | | | | | | | | | | |
| *µCT* | 8.30 | 8.31 | 8.31 | 8.31 | – | 8.27 | 8.29 | 8.40 | 8.32 | – | -0.16% | 1.56% |
| *SL* | 8.40 | 8.43 | 8.44 | 8.42 | 1.40% | 8.35 | 8.41 | 8.44 | 8.40 | 0.96% | 0.28% | 1.07% |
| *SfM* | 8.27 | 8.28 | 8.28 | 8.28 | -0.36% | 8.12 | 8.12 | 8.14 | 8.13 | -2.32% | 1.85% | 1.95% |
| *Sphaeroptica* | 8.19 | 8.19 | 8.29 | 8.22 | -1.00% | 8.20 | 8.21 | 8.25 | 8.22 | -1.20% | 0.04% | 1.22% |

**Table 2. Average of the differences for all measurements for *Eupholus schoenherri*.**

| | Average difference A | Average difference B | Average difference A & B |
|---|---|---|---|
| **SL** | 0.96% | 0.13% | 0.54% |
| **SfM** | -2.02% | -2.35% | -2.19% |
| **Sphaeroptica** | -2.29% | -2.43% | -2.36% |

*Lycus rostratus*. Four distances were measured for Lycus rostratus. All these measurements are comparable between µCT, SfM and Sphaeroptica (Tables 3 and 4). The inter-operator variability is negligible. The biggest difference is with the SL model on the L1-R1 and L2-R2 landmarks that are located on sharp-pointed ends (Fig 19). These features appear to be more smoothed on the SL model (Fig 15), which explains this difference. For other distances, the measurements are relatively similar. Overall, the measurements from different techniques for *Lycus rostratus* are comparable, showing minor deviations. The SL model exhibits the most significant deviations on specific landmarks due to smoothing effects, which are likely present

**Table 3. Landmark measurements (in mm) for *Lycus rostratus*.**

| | Operator A | | | | | Operator B | | | | | A-B | A and B |
|---|---|---|---|---|---|---|---|---|---|---|---|---|
| | A1 min | A2 | A3 max | Avg | Diff µCT | B1 min | B2 | B3 max | Avg | Diff µCT | Diff | (MAX-MIN) / AVG |
| **F-B** | | | | | | | | | | | | |
| µCT | 14.85 | 14.86 | 14.91 | 14.87 | – | 14.91 | 14.91 | 14.92 | 14.91 | – | -0.27% | 0.47% |
| SL | 14.83 | 14.84 | 14.95 | 14.87 | 0.00% | 14.88 | 14.91 | 14.93 | 14.91 | -0.04% | -0.22% | 0.81% |
| SfM | 14.74 | 14.75 | 14.76 | 14.75 | -0.83% | 14.73 | 14.73 | 14.74 | 14.73 | -1.21% | 0.11% | 0.20% |
| Sphaeroptica | 14.77 | 14.77 | 14.78 | 14.77 | -0.67% | 14.73 | 14.73 | 14.75 | 14.74 | -1.18% | 0.25% | 0.34% |
| **L1-R1** | | | | | | | | | | | | |
| µCT | 5.86 | 5.86 | 5.86 | 5.86 | – | 5.86 | 5.87 | 5.87 | 5.87 | – | -0.11% | 0.17% |
| SL | 5.42 | 5.46 | 5.47 | 5.45 | -7.00% | 5.47 | 5.48 | 5.49 | 5.48 | -6.59% | -0.55% | 1.28% |
| SfM | 5.75 | 5.77 | 5.78 | 5.77 | -1.59% | 5.75 | 5.77 | 5.79 | 5.77 | -1.65% | -0.06% | 0.69% |
| Sphaeroptica | 5.77 | 5.78 | 5.82 | 5.79 | -1.19% | 5.84 | 5.84 | 5.85 | 5.84 | -0.40% | -0.91% | 1.38% |
| **L2-R2** | | | | | | | | | | | | |
| µCT | 4.49 | 4.50 | 4.51 | 4.50 | – | 4.54 | 4.56 | 4.57 | 4.56 | – | -1.24% | 1.77% |
| SL | 4.11 | 4.20 | 4.28 | 4.20 | -6.74% | 4.23 | 4.27 | 4.27 | 4.26 | -6.58% | -1.41% | 4.02% |
| SfM | 4.35 | 4.35 | 4.38 | 4.36 | -3.11% | 4.32 | 4.35 | 4.40 | 4.36 | -4.39% | 0.08% | 1.84% |
| Sphaeroptica | 4.45 | 4.45 | 4.45 | 4.45 | -1.11% | 4.45 | 4.47 | 4.48 | 4.47 | -1.98% | -0.37% | 0.67% |
| **V2-V3** | | | | | | | | | | | | |
| µCT | 5.41 | 5.42 | 5.42 | 5.42 | – | 5.41 | 5.41 | 5.41 | 5.41 | – | 0.12% | 0.18% |
| SL | 5.50 | 5.52 | 5.52 | 5.51 | 1.78% | 5.41 | 5.42 | 5.56 | 5.46 | 0.99% | 0.92% | 2.73% |
| SfM | 5.41 | 5.42 | 5.43 | 5.42 | 0.06% | 5.46 | 5.46 | 5.48 | 5.47 | 1.05% | -0.85% | 1.29% |
| Sphaeroptica | 5.39 | 5.40 | 5.41 | 5.40 | -0.31% | 5.40 | 5.41 | 5.45 | 5.42 | 0.18% | -0.37% | 1.11% |

**Table 4. Average of the differences for all measurements for *Lycus rostratus*.**

| | Average difference A | Average difference B | Average difference A & B |
|---|---|---|---|
| **SL** | -2.99% | -3.06% | -3.02% |
| **SfM** | -1.37% | -1.55% | -1.46% |
| **Sphaeroptica** | -0.82% | -0.84% | -0.83% |

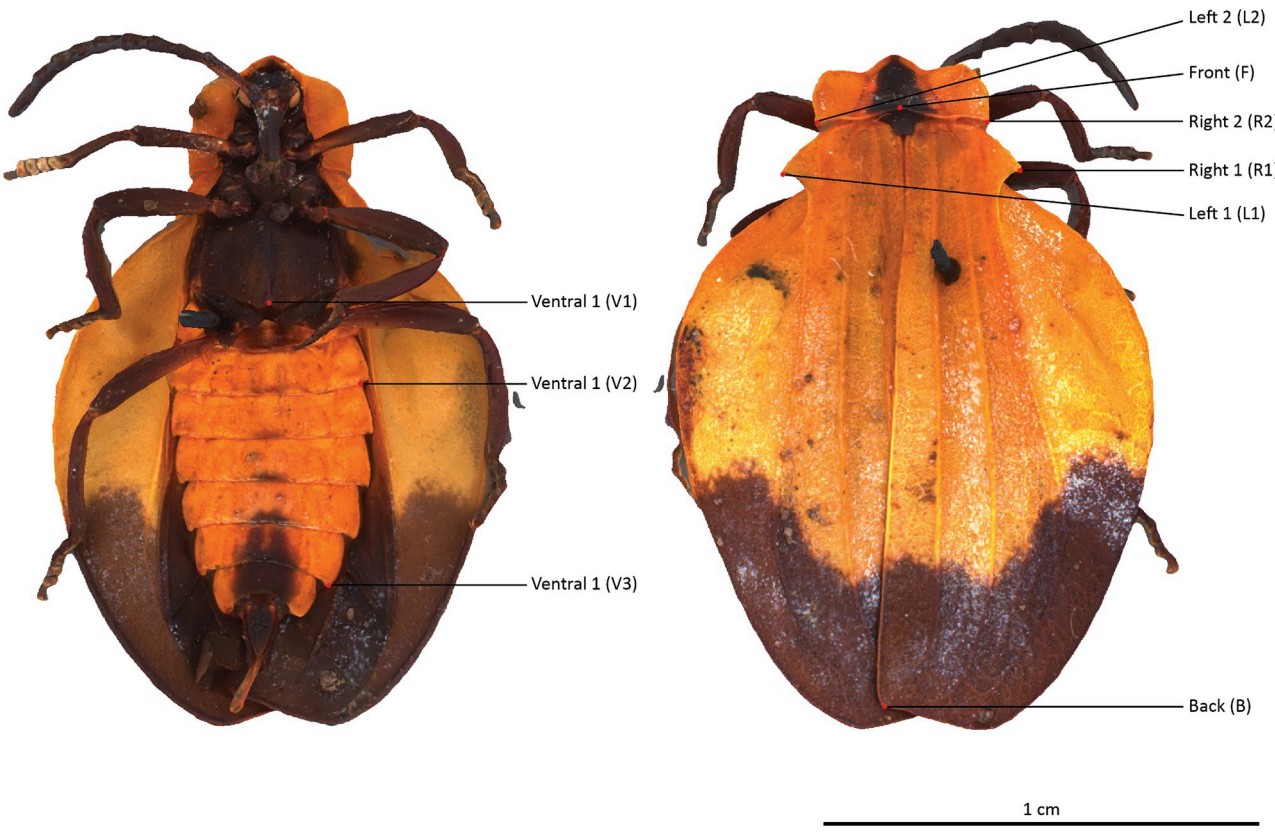

**Fig 19. Lycus rostratus.** Landmarks position on the SfM model.

to a lesser extent in SfM. However, inter-operator variability remains minimal across all techniques. Sphaeroptica exhibits an average difference of less than 1% (Table 4).

*Philodicus* sp. As with *Lycus rostratus*, four distances were measured (Fig 20, Table 5). All measurements are comparable, with an average difference compared to μCT not exceeding 1.2% (Table 6). An error of 3% was observed for Operator A when measuring the distance between F-B on the SL model, while the error was only 0.2% for Operator B. This suggests an error in landmark positioning rather than a real model difference (Fig 20, Table 5).

*Ovalisia* **sp**. All the measurements are comparable on this specimen, with a difference compared to μCT that is not bigger than 1.2% (Fig 21, Tables 7 and 8). In this specimen, the inter-operator variability is very low.

The average absolute difference per technique for all measurements, users, and specimens showed that, on average, SL is the most different from μCT, likely due to a smoother and a less detailed mesh. SfM and Sphaeroptica results are very close to μCT, with differences around 1%. The cumulated differences with μCT confirm that Sphaeroptica is the closest measurement workflow to the μCT one used as reference (Table 9).

The average values are close between the different models and/or techniques. Sphaeroptica seems to be the best alternative to μCT for accurate measurements. Nevertheless, the small number of individual measurements (n = 2x3) does not allow us to perform statistical tests.

In order to better evaluate the dispersion of the measurements for all techniques and how Sphaeroptica is a suitable workflow for geometric morphometrics analyses (GM), operator B

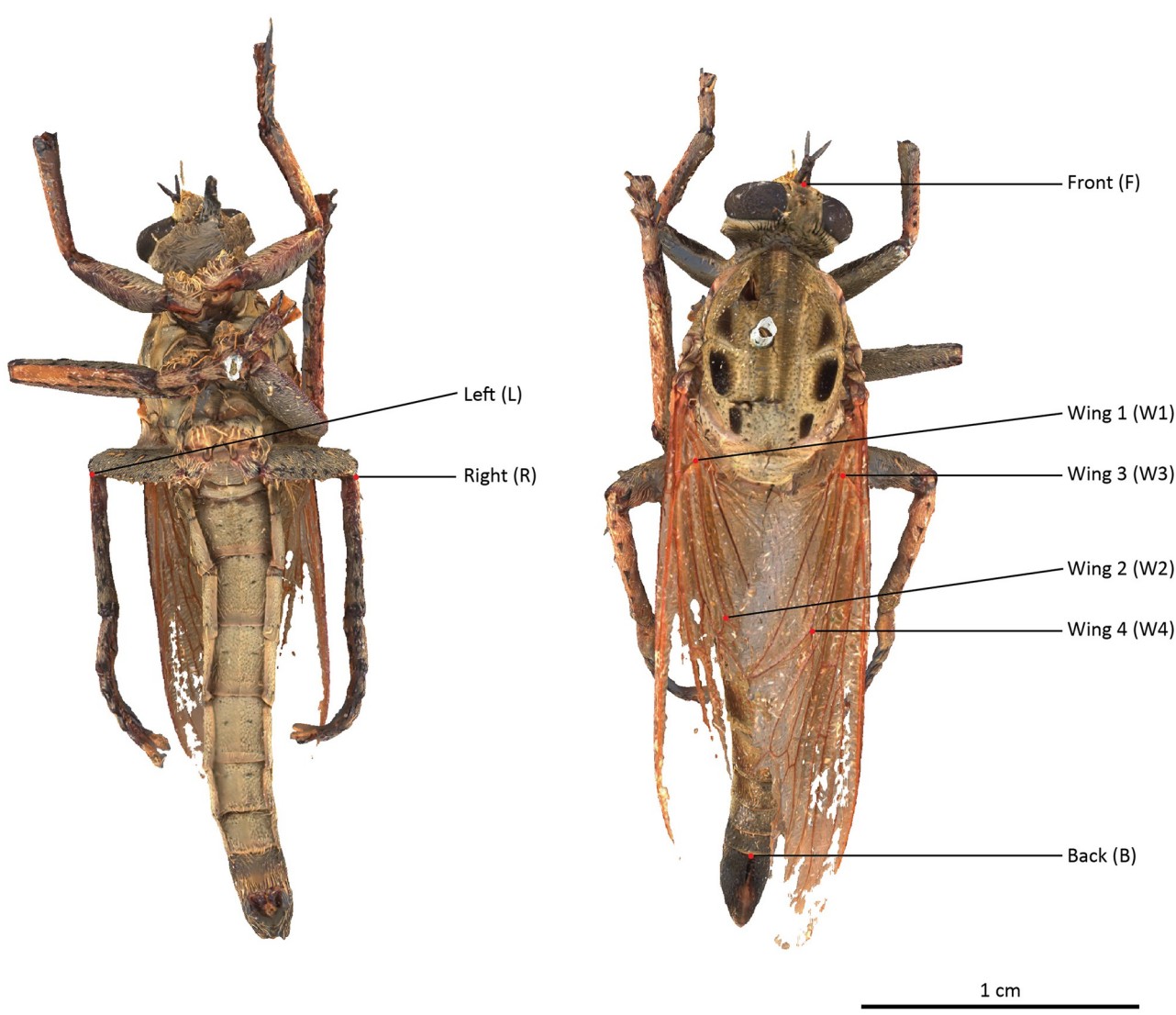

**Fig 20. *Philodicus* sp. Landmarks position on the SfM model.**

measured the L-R distance of the *Ovalisia* sp. specimen 20 times for each technique, as usually performed in GM studies (Table 10).

The distributions of the 20 measurements of each technique are homogenous (Table 11, Fig 22) as demonstrated by the alpha value of repeatability test of Cronbach using PSPP package (Cronbach's Alpha = 1).

The SfM measurements present the smallest dispersion (0.37%). This can be explained by the initial choice of the landmarks. This choice was achieved on the SfM model by selecting the reference points that are the easiest to position using this technique, since selecting features on the μCT would sometimes be difficult to find on the other models due to the difference of resolution and absence of texture. The Sphaeroptica values show the largest dispersion (1.03%). It must be noticed that the SfM and Sphaeroptica measurements are both derived from the same set of oriented images. This difference can be explained by the fact that the positioning of the landmarks using Sphaeroptica is not restricted to the surface of the 3D model,

**Table 5. Landmark measurements (in mm) for *Philodicus sp*.**

| | Operator A | | | | | Operator B | | | | | A-B | A and B |
|---|---|---|---|---|---|---|---|---|---|---|---|---|
| | A1 min | A2 | A3 max | Avg | Diff μCT | B1 min | B2 | B3 max | Avg | Diff μCT | Diff | (MAX-MIN) / AVG |
| **F-B** | | | | | | | | | | | | |
| *μCT* | 24.12 | 24.14 | 24.15 | 24.14 | – | 24.16 | 24.17 | 24.18 | 24.17 | – | -0.14% | 0.25% |
| *SL* | 24.08 | 25.22 | 25.25 | 24.85 | 2.96% | 24.19 | 24.20 | 24.24 | 24.21 | 0.17% | 2.64% | 4.77% |
| *SfM* | 24.13 | 24.14 | 24.15 | 24.14 | 0.01% | 24.14 | 24.14 | 24.28 | 24.19 | 0.07% | -0.19% | 0.62% |
| *Sphaeroptica* | 24.13 | 24.14 | 24.15 | 24.14 | 0.01% | 24.12 | 24.14 | 24.14 | 24.13 | -0.15% | 0.03% | 0.12% |
| **L-R** | | | | | | | | | | | | |
| *μCT* | 9.97 | 9.99 | 10.00 | 9.99 | – | 9.99 | 10.01 | 10.02 | 10.01 | – | -0.20% | 0.50% |
| *SL* | 9.92 | 10.05 | 10.07 | 10.01 | 0.27% | 9.91 | 9.95 | 10.02 | 9.96 | -0.47% | 0.54% | 1.60% |
| *SfM* | 10.14 | 10.17 | 10.18 | 10.16 | 1.77% | 10.02 | 10.03 | 10.14 | 10.06 | 0.57% | 0.99% | 1.58% |
| *Sphaeroptica* | 10.04 | 10.05 | 10.19 | 10.09 | 1.07% | 10.02 | 10.09 | 10.13 | 10.08 | 0.73% | 0.13% | 1.69% |
| **W1-W2** | | | | | | | | | | | | |
| *μCT* | 4.75 | 4.75 | 4.77 | 4.76 | – | 4.71 | 4.73 | 4.77 | 4.74 | – | 0.42% | 1.26% |
| *SL* | 4.81 | 4.83 | 4.86 | 4.83 | 1.61% | 4.80 | 4.82 | 4.90 | 4.84 | 2.18% | -0.14% | 2.07% |
| *SfM* | 4.81 | 4.82 | 4.83 | 4.82 | 1.33% | 4.76 | 4.81 | 4.83 | 4.80 | 1.34% | 0,42% | 1.46% |
| *Sphaeroptica* | 4.79 | 4.80 | 4.84 | 4.81 | 1.12% | 4.73 | 4.77 | 4.82 | 4.77 | 0.77% | 0.77% | 2.30% |
| **W3-W4** | | | | | | | | | | | | |
| *μCT* | 4.88 | 4.91 | 4.92 | 4.90 | – | 4.85 | 4.86 | 4.90 | 4.87 | – | 0.68% | 1.43% |
| *SL* | 4.88 | 4.92 | 4.93 | 4.91 | 0.14% | 4.87 | 4.89 | 4.93 | 4.90 | 0.55% | 0.27% | 1.22% |
| *SfM* | 4.94 | 4.97 | 4.99 | 4.97 | 1.29% | 4.95 | 4.99 | 5.05 | 5.00 | 2.60% | -0.60% | 2.21% |
| *Sphaeroptica* | 4.88 | 4.93 | 4.94 | 4.92 | 0.27% | 4.84 | 4.86 | 4.97 | 4.89 | 0.41% | 0.55% | 2.65% |

**Table 6. Average of the differences for all measurements for *Philodicus* sp.**

| | Average difference A | Average difference B | Average difference A & B |
|---|---|---|---|
| **SL** | 1.24% | 0.61% | 0.92% |
| **SfM** | 1.10% | 1.14% | 1.12% |
| **Sphaeroptica** | 0.62% | 0.44% | 0.53% |

which is reflected by a slightly higher dispersion of the values. Nevertheless, the average value (5.51) is closer to the μCT reference (5.50) than the 2 other techniques (5.59 and 5.47).

## Evaluation of the human and computer times required by the workflows

The following section compares the human time and the computer time required by the different workflows. The time required for specimen manipulation by the operator is not considered in this evaluation, as it is almost the same for the different workflows.

The training time required to master the creation and use of a Sphaeroptica project is less than an hour. In our case, as mentioned above, we use scAnt for image acquisition, and so the evaluation of data acquisition time for Sphaeroptica will be investigated with this configuration.

Concerning the acquisition time, the fastest setup is the μCT with an acquisition time varying between 8 and 25 minutes, depending on the resolution and the μCT scanner (Tables 12–15). The SL ARTEC Micro scanner required about 30 minutes of scan (15 minutes per side) with 1 minute of human time required to change the orientation of the specimen (Tables 12–15). Sphaeroptica and photogrammetry workflows used the scAnt scanner. For Eupholus, 180

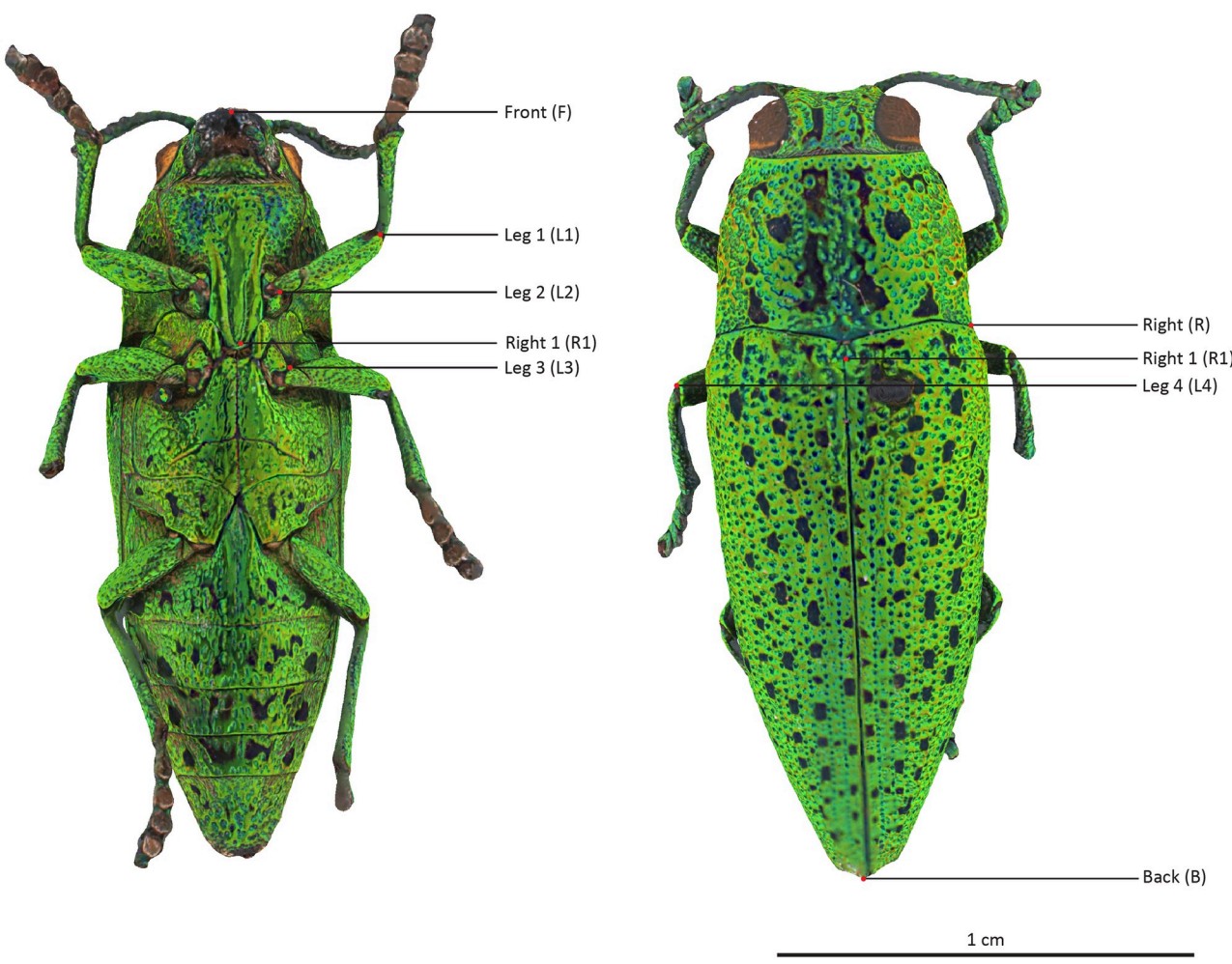

**Fig 21. *Ovalisia* sp. Landmarks position on the Sfm model.**

views corresponding to 5940 pictures were acquired in 2 hours (Table 12). For the 3 other specimens, 180 views corresponding to 2880 pictures required about 1 hour. The acquisition is fully automated and no human time is required during the process (Tables 13–15). When using a setup like scAnt, very little photogrammetry knowledge is necessary to master the acquisition process.

The processing of the 3D model and the texture of the SL workflow using Artec studio is the fastest.

The processing of the µCT data can be divided between the reconstruction of the planar images on the base of the radial X-rays, which is mainly computer time, and the manual segmentation of the images to extract the 3D surface of the specimen. This latter step can be time-consuming since the time to segment the specimen can vary between 8 minutes for the Eupholus to 90 minutes for the *Philodicus* sp. because it displays complex and thin structures.

The shared processing of the Sphaeroptica and photogrammetry workflows is the longest. It could be divided into several phases corresponding to the stacking of the images and to the bundle adjustment. The first phase consists in aligning and stacking the pictures to produce the EOF images. This is the longest step and requires between 4 and 6 hours to produce the

**Table 7. Landmark measurements (in mm) for *Ovalisia* sp.**

| | Operator A | | | | | Operator B | | | | | A-B | A and B |
|---|---|---|---|---|---|---|---|---|---|---|---|---|
| | A1 min | A2 | A3 max | Avg | Diff μCT | B1 min | B2 | B3 max | Avg | Diff μCT | Diff | (MAX-MIN) / AVG |
| **F-B** | | | | | | | | | | | | |
| *μCT* | 17.37 | 17.94 | 17.96 | 17.76 | – | 17.95 | 17.95 | 17.95 | 17.95 | – | -1.08% | 3.30% |
| *SL* | 17.88 | 17.93 | 17.94 | 17.92 | 0.90% | 17.93 | 17.94 | 17.94 | 17.94 | -0.07% | -0.11% | 0.33% |
| *SfM* | 17.89 | 17.89 | 17.89 | 17.89 | 0.75% | 17.90 | 17.90 | 17.91 | 17.90 | -0.26% | -0.07% | 0.11% |
| *Sphaeroptica* | 17.89 | 17.92 | 17.98 | 17.93 | 0.98% | 17.92 | 17.96 | 17.98 | 17.95 | 0.02% | -0.13% | 0.50% |
| **L-R** | | | | | | | | | | | | |
| *μCT* | 5.51 | 5.52 | 5.52 | 5.52 | – | 5.49 | 5.49 | 5.49 | 5.49 | – | 0.49% | 0.55% |
| *SL* | 5.36 | 5.46 | 5.57 | 5.46 | -0.97% | 5.56 | 5.57 | 5.59 | 5.57 | 1.52% | -1.97% | 4.17% |
| *SfM* | 5.46 | 5.46 | 5.48 | 5.47 | -0.91% | 5.47 | 5.47 | 5.47 | 5.47 | -0.36% | -0.06% | 0.37% |
| *Sphaeroptica* | 5.51 | 5.53 | 5.58 | 5.54 | 0.42% | 5.51 | 5.53 | 5.54 | 5.53 | 0.67% | 0.24% | 1.27% |
| **U-D** | | | | | | | | | | | | |
| *μCT* | 4.18 | 4.18 | 4.19 | 4.18 | – | 4.18 | 4.18 | 4.19 | 4.18 | – | 0.00% | 0.24% |
| *SL* | 4.33 | 4.38 | 4.37 | 4.36 | 4.22% | 4.24 | 4.25 | 4.26 | 4.25 | 1.59% | 2.59% | 3.25% |
| *SfM* | 4.18 | 4.18 | 4.18 | 4.18 | -0.08% | 4.18 | 4.19 | 4.20 | 4.19 | 0.16% | -0.24% | 0.48% |
| *Sphaeroptica* | 4.14 | 4.17 | 4.18 | 4.16 | -0.48% | 4.17 | 4.18 | 4.18 | 4.18 | -0.16% | -0.32% | 0.96% |
| **F-U** | | | | | | | | | | | | |
| *μCT* | 6.77 | 6.77 | 6.79 | 6.78 | – | 6.76 | 6.75 | 6.76 | 6.76 | – | 0.30% | 0.59% |
| *SL* | 6.78 | 6.81 | 6.82 | 6.80 | 0.39% | 6.81 | 6.83 | 6.83 | 6.82 | 0.99% | -0.29% | 0.73% |
| *SfM* | 6.74 | 6.74 | 6.76 | 6.75 | -0.44% | 6.76 | 6.77 | 6.77 | 6.77 | 0.15% | -0.30% | 0.44% |
| *Sphaeroptica* | 6.76 | 6.76 | 6.77 | 6.76 | -0.20% | 6.74 | 6.78 | 6.81 | 6.78 | 0.30% | -0.20% | 1.03% |
| **B-U** | | | | | | | | | | | | |
| *μCT* | 12.34 | 12.34 | 12.36 | 12.35 | – | 12.33 | 12.34 | 12.34 | 12.34 | – | 0.08% | 0.24% |
| *SL* | 12.27 | 12.28 | 12.32 | 12.29 | -0.46% | 12.26 | 12.29 | 12.30 | 12.28 | -0.43% | 0.05% | 0.49% |
| *SfM* | 12.31 | 12.32 | 12.33 | 12.32 | -0.22% | 12.30 | 12.30 | 12.29 | 12.30 | -0.32% | 0.19% | 0.32% |
| *Sphaeroptica* | 12.29 | 12.32 | 12.37 | 12.33 | -0.16% | 12.30 | 12.32 | 12.32 | 12.31 | -0.19% | 0.11% | 0.65% |
| **L1-L2** | | | | | | | | | | | | |
| *μCT* | 2.93 | 2.93 | 2.94 | 2.93 | – | 2.90 | 2.93 | 2.93 | 2.92 | – | 0.46% | 1.37% |
| *SL* | 2.98 | 3.01 | 3.03 | 3.01 | 2.50% | 3.02 | 3.02 | 3.03 | 3.02 | 3.54% | -0.55% | 1.66% |
| *SfM* | 3.00 | 3.01 | 3.01 | 3.01 | 2.50% | 2.92 | 2.94 | 2.96 | 2.94 | 0.68% | 2.27% | 3.03% |
| *Sphaeroptica* | 2.98 | 3.01 | 3.03 | 3.01 | 2.50% | 2.94 | 2.95 | 2.95 | 2.95 | 0.91% | 2.04% | 3.02% |
| **L3-L4** | | | | | | | | | | | | |
| *μCT* | 2.90 | 2.90 | 2.99 | 2.93 | – | 2.93 | 2.94 | 2.94 | 2.94 | – | -0.23% | 3.07% |
| *SL* | 2.95 | 2.96 | 2.97 | 2.96 | 1.02% | 2.95 | 2.97 | 2.98 | 2.97 | 1.02% | -0.22% | 1.01% |
| *SfM* | 2.93 | 2.95 | 2.96 | 2.95 | 0.57% | 2.92 | 2.92 | 2.92 | 2.92 | -0.57% | 0.91% | 1.36% |
| *Sphaeroptica* | 2.91 | 3.02 | 3.04 | 2.99 | 2.05% | 2.95 | 2.96 | 2.97 | 2.96 | 0.79% | 1.01% | 4.37% |

**Table 8. Average of the differences for all measurements for *Ovalisia* sp.**

| | Average difference A | Average difference B | Average difference A & B |
|---|---|---|---|
| **SL** | 1.09% | 1.16% | 1.13% |
| **SfM** | 0.31% | -0.07% | 0.12% |
| **Sphaeroptica** | 0.73% | 0.33% | 0.53% |

**Table 9. Absolute average difference and cumulated differences per technique for all specimens, all users and all landmarks.**

|  | Absolute average differences | Cumulated differences with μCT |
|---|---|---|
| **SL** | 1.74% | 62.49% |
| **SfM** | 1.22% | 43.95% |
| **Sphaeroptica** | 0.99% | 35.57% |

180 stacked views. It seems that the Open Source Enfuse stacking software used by the scAnt scanner is not using the GPU, which could probably be improved. Preliminary tests comparing Enfuse, Zerene stacker, and Helicon show that the time to create one stacked image from the 33 raw pictures of Eupholus requires 120 seconds with Enfuse, 60 seconds with Zerene, and 8 seconds with Helicon focus as it uses the NVIDIA GPU (GeForce RTX 3070) (Fig 23). Using Helicon Focus for the stacking step could reduce the time for this part of the workflow to 26 minutes instead of 6 hours.

Agisoft Metashape required about 3 minutes for the bundle adjustment of the 180 views. This is the end step for the Sphaeroptica workflow (Tables 12–15).

For the SfM workflow, the calculation of the dense cloud, the mesh and the texture by Agisoft Metashape requires about 20-30 additional minutes of computer time (Tables 12–15).

## Discussion and conclusion

This paper presents a novel method for the digitization of entomological specimens, enabling pseudo 3D viewing of the specimen with 3D landmarks positioning and export as CSV file. Sphaeroptica allows precise linear measurements directly in the application. We reviewed this workflow in comparison with other standard 3D approaches such as SfM, SL and μCT.

**Table 10. L-R distances of the *Ovalisia* sp. (in mm).**

| L-R |  |  |  |  |  |  |  |  |  |  |
|---|---|---|---|---|---|---|---|---|---|---|
|  | *1* | *2* | *3* | *4* | *5* | *6* | *7* | *8* | *9* | *10* |
| μ-CT | 5.48 | 5.51 | 5.51 | 5.50 | 5.50 | 5.49 | 5.49 | 5.51 | 5.51 | 5.50 |
| SL | 5.60 | 5.60 | 5.62 | 5.57 | 5.60 | 5.59 | 5.58 | 5.62 | 5.61 | 5.59 |
| SfM | 5.46 | 5.46 | 5.48 | 5.48 | 5.48 | 5.48 | 5.46 | 5.47 | 5.46 | 5.46 |
| Sphaeroptica | 5.50 | 5.50 | 5.52 | 5.52 | 5.51 | 5.50 | 5.51 | 5.49 | 5.52 | 5.51 |
|  | *11* | *12* | *13* | *14* | *15* | *16* | *17* | *18* | *19* | *20* |
| μ-CT | 5.48 | 5.50 | 5.48 | 5.49 | 5.49 | 5.50 | 5.48 | 5.51 | 5.49 | 5.49 |
| SL | 5.59 | 5.59 | 5.58 | 5.59 | 5.57 | 5.59 | 5.58 | 5.62 | 5.59 | 5.58 |
| SfM | 5.46 | 5.47 | 5.47 | 5.47 | 5.47 | 5.47 | 5.46 | 5.48 | 5.47 | 5.47 |
| Sphaeroptica | 5.51 | 5.50 | 5.55 | 5.53 | 5.51 | 5.51 | 5.52 | 5.53 | 5.54 | 5.49 |

**Table 11. L-R distances of the *Ovalisia* sp. specimen distribution parameters of the measurements.**

| L-R | Avg | Stdev | Diff μCT | (MAX-MIN) / AVG | 95% confidence |
|---|---|---|---|---|---|
| **μ-CT** | 5.50 | 0.01 | – | 0.69% | 5.47—5.52 |
| **SL** | 5.59 | 0.01 | 1.77% | 0.92% | 5.56—5.62 |
| **SfM** | 5.47 | 0.01 | -0.51% | 0.37% | 5.45—5.48 |
| **Sphaeroptica** | 5.51 | 0.02 | 0.33% | 1.03% | 5.48—5.55 |

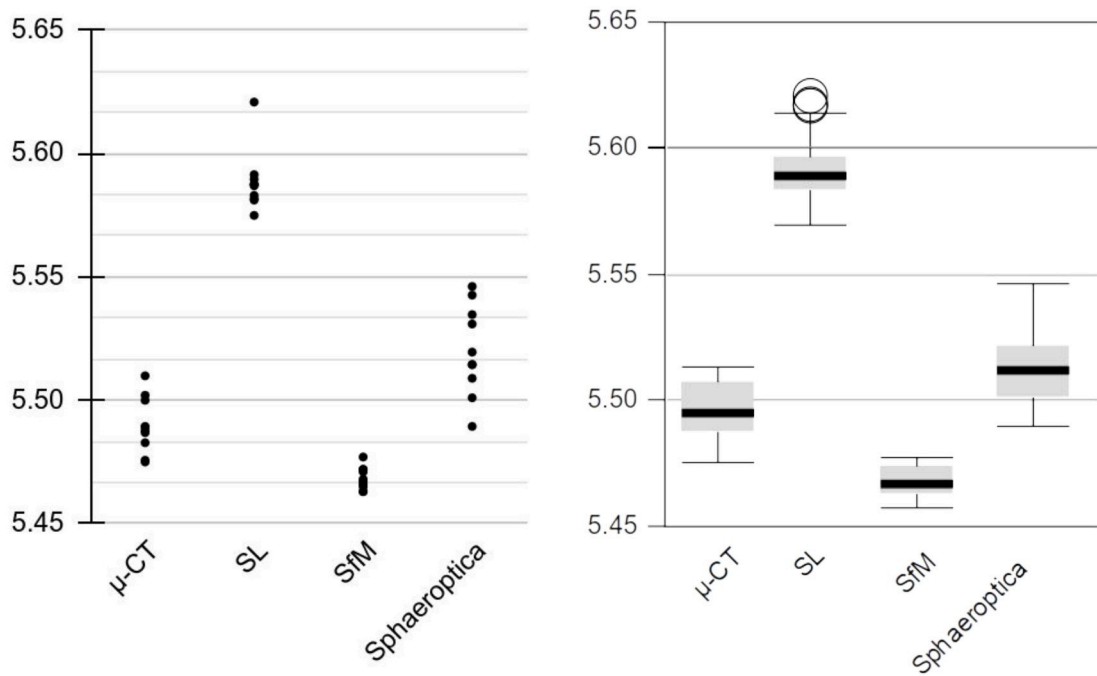

**Fig 22. Dispersion of the L-R distance of the _Ovalisia_ sp. specimen (in mm).**

**Table 12. _Eupholus schoenherri_, computer time (PT) and human time (HT) required for the different workflows.**

| _Eupholus schoenherri_ | Sphaeroptica | Photogrammetry | SL | μCT |
|---|---|---|---|---|
| | **Capture 180 views / 5940 pictures** <br> 2h PT | | **Acquisition** <br> 30m PT <br> 01m HT | **Acquisition** <br> 08m PT |
| | **Stacking** <br> Enfuse no GPU: 6h PT <br> Helicon Graphic GPU: 26m PT | | **Processing** <br> 17m PT <br> 02m HT | **Reconstruction** <br> 10m PT |
| | **Bundle adjustment** <br> 03m PT <br> 02m HT | | | |
| | **No 3D model** | **3D Modeling** <br> 26m PT <br> Mesh ultra-high, moderate filtering | | **Segmentation** <br> 15m HT |
| | **Native texture** | **Texture** <br> 8192px x 8192px <br> 02m PT | **Texture** <br> 8192px x 8192px <br> 03m PT | **No texture** |
| **Total PT** | 8h06m Enfuse <br> 2h29m Helicon | 8h31m Enfuse <br> 2h57m Helicon | 50m | 18m |
| **Total HT** | 02m | 02m | 03m | 15m |
| **Size of raw data** | 169,8 GB tiff + jpg | 169,8 GB | 7.4 GB (raw) | 9.63 GB |
| **Size of processed data** | 206 MB jpg | 33 MB (obj + mtl + jpg) | 17 MB (obj + mtl + jpg) | 392 MB (stl) |

The time for the manipulation of the specimen and the measures are not included as not different between workflows.

**Table 13.** *Lycus rostratus*, computer time (PT) and human time (HT) required for the different workflows.

| Lycus rostratus | Sphaeroptica | Photogrammetry | SL | μCT |
|---|---|---|---|---|
| | **Capture 180 views / 2880 pictures**<br>1h PT | | **Acquisition**<br>30m PT<br>01m HT | **Acquisition**<br>22m PT |
| | **Stacking**<br>Enfuse no GPU: 4h PT<br>Helicon Graphic GPU: 21m PT | | **Processing**<br>17m PT<br>02m HT | **Reconstruction**<br>10m PT |
| | **Bundle adjustment**<br>03m PT<br>02m HT | | | |
| | **No 3D model** | **3D Modeling**<br>19m PT<br>Mesh ultra-high, moderate filtering | | **Segmentation**<br>30m HT |
| | **Native texture** | **Texture**<br>8192px x 8192px<br>02m PT | **Texture**<br>8192px x 8192px<br>03m PT | **No texture** |
| **Total PT** | 5h03m Enfuse<br>1h24m Helicon | 1h24m Enfuse<br>1h45m Helicon | 50m | 32m |
| **Total HT** | 02m | 02m | 03m | 30m |
| **Size of raw data** | 87,3 GB tiff + jpg | 87,3 GB | 8,6 GB (raw) | 9.34 GB |
| **Size of processed data** | 387 MB jpg | 16 MB (obj + mtl + jpg) | 36 MB (obj + mtl + jpg) | 1 GB (stl) |

The time for the manipulation of the specimen and the measures are not included as not different between workflows.

Sphaeroptica enables measurements with an average deviation close to 1% compared to the μCT model, making Sphaeroptica a competitive tool for the metric study of small and detailed specimens.

Sphaeroptica is the only method able to visualize the details of the wings and/or setae correctly using native color images (Figs 12 and 16, Table 16). Even μCT poses challenges, given

**Table 14.** *Ovalisia* sp., computer time (PT) and human time (HT) required for the different workflows.

| Ovalisia sp. | Sphaeroptica | Photogrammetry | SL | μCT |
|---|---|---|---|---|
| | **Capture 180 views / 2880 pictures**<br>1h PT | | **Acquisition**<br>30m PT<br>01m HT | **Acquisition**<br>16m PT |
| | **Stacking**<br>Enfuse no GPU: 4h PT<br>Helicon Graphic GPU: 21m PT | | **Processing**<br>17m PT<br>02m HT | **Reconstruction**<br>10m PT |
| | **Bundle adjustment**<br>03m PT<br>02m HT | | | |
| | **No 3D model** | **3D Modeling**<br>17m PT<br>Mesh ultra-high, moderate filtering | | **Segmentation**<br>15m HT |
| | **Native texture** | **Texture**<br>8192px x 8192px<br>02m PT | **Texture**<br>8192px x 8192px<br>05m PT | **No texture** |
| **Total PT** | 5h03m Enfuse<br>4h24m Helicon | 5h22m Enfuse<br>4h43m Helicon | 52m | 26m |
| **Total HT** | 02m | 02m | 03m | 15m |
| **Size of raw data** | 90,3 GB tiff + jpg | 90,3 GB | 4,72 GB (raw) | 9 GB |
| **Size of processed data** | 546 MB jpg | 14 MB (obj + mtl + jpg) | 36 MB (obj + mtl + jpg) | 486 MB (stl) |

The time for the manipulation of the specimen and the measures are not included as not different between workflows.

**Table 15. *Philodicus* sp., computer time (CT) and human time (HT) required for the different workflows.**

| *Philodicus* sp. | Sphaeroptica | Photogrammetry | SL | μCT |
|---|---|---|---|---|
| | **Capture 180 views / 2880 pictures** 1h PT | | **Acquisition** 40m PT 01m HT | **Acquisition** 25m PT |
| | **Stacking** Enfuse no GPU: 5h PT Helicon Graphic GPU: 23m PT | | **Processing** 20m PT 03m HT | **Reconstruction** 10m PT |
| | **Bundle adjustment** 03m PT 02m HT | | | |
| | **No 3D model** | **3D Modeling** 21 min PT Mesh ultra-high, moderate filtering | | **Segmentation** 60-90m HT |
| | **Native texture** | **Texture** 8192px x 8192px 04m PT | **Texture** 8192px x 8192px 05m PT | **No texture** |
| **Total PT** | 6h03m Enfuse 1h26m Helicon | 5h28m Enfuse 1h51m Helicon | 1h05m | 35m |
| **Total HT** | 02m | 02m | 04m | 60-90m |
| **Size of raw data** | 91.1GB tiff + jpg | 91.1 GB | 4.3 GB (raw) | 12.1 GB |
| **Size of processed data** | 455 MB jpg | 34 MB (obj + mtl + jpg) | 51 MB (obj + mtl + jpg) | 294 MB (stl) |

The time for the manipulation of the specimen and the measures are not included as not different between workflows.

the low density and the complexity of rendering such delicate structures in 3D. Sphaeroptica enables good visualization and measurements on wings without the need to detach them from the body [8], thus it is less invasive for the specimens.

Sphaeroptica excels in rendering the most accurate color information. While Structure from Motion (SfM) comes close in terms of color fidelity, it slightly sacrifices sharpness and

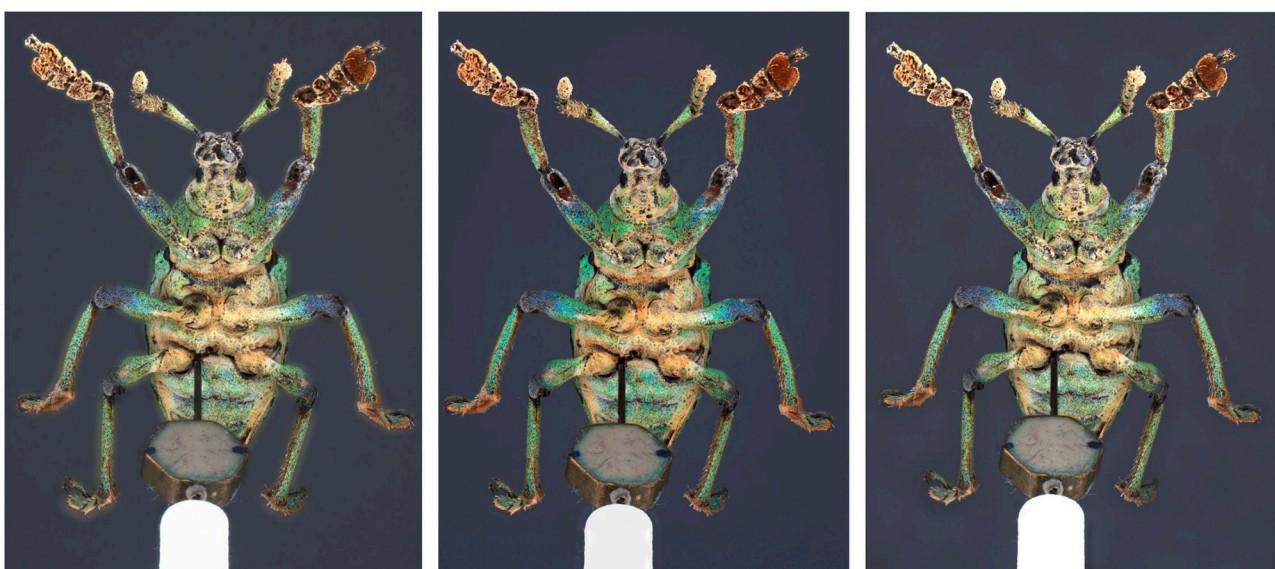

**Fig 23. Result of stacked image with 3 different softwares.** Stacked images produced by Enfuse (120 s), Zerene Stacker (60 s) and Helicon Focus (8 s). See the hexagonal bead attached to the pin between the holder and the specimen.

**Table 16. Key points of the different techniques.**

|  | μCT | SL | SfM | Sphaeroptica |
|---|---|---|---|---|
| **3D** | 3D from segmentation | Direct 3D | Computed 3D | Multiview pseudo 3D |
|  | STL | OBJ | OBJ | TIFF, JPG |
| **Color texture** | No | Yes | Computed Photorealistic | Photorealistic |
| **Visualization of small details** | Variable | No | No | Yes |
| **Time of acquisition** | **30 minutes** | 30 minutes | 1-2 hours | 1-2 hours |
| **Time of processing** | 15-90 minutes | <30 minutes | 90-100 minutes (GPU) 6-7 hours (no GPU) | 30-45 minutes (GPU) 5-6 hours (no GPU) |
| **Cost of equipment** | >100 k€ | >10 k€ | <10 k€ | <10 k€ |

details due to texture blending. In contrast, μCT lacks the ability to capture texture altogether (Table 16).

From a cost perspective, both Sphaeroptica and SfM provide the cheapest solutions, whereas SL and μCT equipment incur significantly higher expenses (Table 16).

In terms of time efficiency, SL emerges as the quickest, followed by μCT (Table 10). Nevertheless, the human processing time for μCT (image segmentation) varies depending on specimen complexity, with some specimens allowing fast automatic segmentation, while others necessitate time-consuming manual segmentation. Sphaeroptica and SfM prove to be less time-efficient, primarily due to the time required by Enfuse in the stacking process.

We think that it is still possible to improve the Sphaeroptica workflow. The predominant metric error can indeed be attributed to the scaling discrepancies and/or stacking deformation. Consequently, the subsequent phases of this research will focus on identifying an improved scaling solution.

Even though the pictures are very good, it is possible to further improve the quality and the definition of the pictures using DSLR/Mirrorless cameras with high quality macro lenses instead of an industrial camera with HD C-Mount lens. The scAnt hardware used for the acquisition can also operate with these cameras.

Enfuse is not using GPU during the stacking process. The test with Helicon focus using the GPU drastically reduces the time devoted to the stacking step by a factor 15X using a NVIDIA GeForce RTX3070. Open Source packages using GPU have to be evaluated in order to speedup the stacking step of the workflow. For larger specimens, the stacking step is not required, making the photogrammetry/Sphaeroptica workflows time efficient.

Shortcomings of Sphaeroptica are similar to the ones of SfM, given their shared 3D alignment method. This may pose difficulties for flat specimens. However, this challenge can be mitigated by incorporating supplementary elements, such as the hexagonal bead utilized for both scaling and alignment purposes. Furthermore, potential errors in alignment and measurements may arise from the movement of wings or legs due to displacement of the specimen during the digitization. Future work includes the improvement of the internal scaling method and the use of a robotic arm in order to turn the camera around the specimen instead of moving the specimen in front of the camera.

While Sphaeroptica workflow has the drawback of being slower than μCT and SL, its affordability, coupled with the automated production of accurate and photorealistic virtual specimens with a high level of visible details make this workflow the most appropriate for the digitization, visualization, and measurement of small natural history specimens with complex shape and detailed texture such as arthropods.

To the best of our knowledge, the proposed system is the only one capable of 360° color rendering of tiny structures such as hairs, while being able to perform 3D measurements. The

results of our quantitative comparison show that Sphaeroptica is an accurate tool to define landmarks on specimens, to perform measurements, and to export the landmarks coordinates to geometric morphometrics packages. This tends to corroborate the results of Olkowicz *et al.* [27] stating that the distortion due to stacking is minimal enough, as the measurement error is close to 1% in relation to our reference standard (μCT). While we tried to place landmarks that can be easily found on all 4 visualizations, it must be said that μCT and, even more, Sphaeroptica, are the only techniques allowing the placements of precise landmarks using small details which are not visible on the SL and SfM models. Sphaeroptica 1.0 does not allow the creation of semi-landmarks but this feature will be considered for future versions of the application.

The use of Sphaeroptica is not limited to small arthropods. Sphaeroptica can be used for objects of any size and the same visualization/measurement workflow could be applied to objects with complex surface structures, such as feathers or straw that can be found in vertebrates and ethnographic collections. It could be also used for geological samples such as gems and crystals, which present reflective and translucent surfaces that are challenging standard 3D digitization methods.

From a technical standpoint, we want to leverage open-source solutions such as COLMAP for the pictures alignment, thereby ensuring the entire process adheres to an open-science framework. We have been using Agisoft software since 2013 as it was the most effective and easy software, and we benefit from educational licenses. Today, Agisoft changed its policy for educational licenses, excluding most of the museums and making Metashape professional very expensive for non-profit organizations.

Furthermore, we aim to enhance the user-friendliness of the interface of Sphaeroptica. In particular, we would like to turn the desktop software as a Web application, ensuring cross-platform compatibility, regardless of their versions.Finally, we aim to integrate Sphaeroptica as a visualization plugin for the Orthanc open-source DICOM server [28] to store and share the Sphaeroptica data according to the DICOM format, a globally recognized standard in medical imaging, which can serve as a container for imaging data and associated metadata encountered in natural history collections.

## Acknowledgments

The authors wish to thank Stephane Hanot and Stefan Kerkhof for providing the specimens used as case studies.

## Author Contributions

**Conceptualization:** Aurore Mathys, Yann Pollet.

**Data curation:** Yann Pollet, Patrick Semal.

**Formal analysis:** Aurore Mathys, Yann Pollet, Patrick Semal.

**Funding acquisition:** Didier Vandenspiegel, Patrick Semal.

**Investigation:** Aurore Mathys, Yann Pollet, Patrick Semal.

**Methodology:** Aurore Mathys, Yann Pollet, Patrick Semal.

**Project administration:** Patrick Semal.

**Resources:** Aurore Mathys, Yann Pollet, Wouter Dekoninck, Didier Vandenspiegel, Patrick Semal.

**Supervision:** Adrien Gressin, Xavier Muth, Didier Vandenspiegel, Sébastien Jodogne, Patrick Semal.

**Validation:** Aurore Mathys, Yann Pollet.

**Visualization:** Aurore Mathys, Yann Pollet, Patrick Semal.

**Writing – original draft:** Aurore Mathys, Patrick Semal.

**Writing – review & editing:** Aurore Mathys, Yann Pollet, Adrien Gressin, Xavier Muth, Jonathan Brecko, Wouter Dekoninck, Didier Vandenspiegel, Sébastien Jodogne, Patrick Semal.

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
