## [Decision Letter · Decision Letter 0]

10 Jun 2024

PONE-D-24-13247Sphaeroptica: A tool for 3D taxonomy and geometric morphometrics on arthropodsPLOS ONE

Dear Dr. Mathys,

Thank you for submitting your manuscript to PLOS ONE. After careful consideration, we feel that it has merit but does not fully meet PLOS ONE’s publication criteria as it currently stands. Therefore, we invite you to submit a revised version of the manuscript that addresses the points raised during the review process.

Based on reviewers’ comments, minor revisions are required in order to improve the manuscript. In particular, some concerns about the error’s calculation were raised and should be addressed.

We look forward to receiving your revised manuscript.

Kind regards,

Giulia Pascoletti, Ph.D.

Academic Editor

PLOS ONE

Journal Requirements:

Reviewers' comments:

Reviewer's Responses to Questions

**Comments to the Author**

1. Is the manuscript technically sound, and do the data support the conclusions?

Reviewer #1: Yes

Reviewer #2: Yes

2. Has the statistical analysis been performed appropriately and rigorously? 

Reviewer #1: Yes

Reviewer #2: N/A

3. Have the authors made all data underlying the findings in their manuscript fully available?

Reviewer #1: Yes

Reviewer #2: Yes

4. Is the manuscript presented in an intelligible fashion and written in standard English?

Reviewer #1: Yes

Reviewer #2: Yes

5. Review Comments to the Author

Reviewer #1: It was a pleasure to read this manuscript introducing "Sphaeroptica," a new tool for 3D analysis of small and complex morphologies, specifically arthropods. Overall, the manuscript is well-written and structured, clearly outlining the benefits of this tool, providing sufficient details on its functionality, comparing it with other scanning techniques, and thoroughly discussing its limitations and potential improvements. The calculated error percentages across techniques are low and within the range of intra- and inter-observer bias, which is also minimal. I agree that, with certain limitations addressed, this tool could significantly benefit the study of similar 3D forms. However, I have a few substantial concerns that should be addressed to improve the manuscript:

1. Although the manuscript emphasizes 3D geometric morphometrics (GM), none of the analyses directly relate to GM. Landmark coordinates are used for linear measurements, but the coordinates were not extracted and superimposed to obtain shape variables. This process could be used to compare different specimens in Procrustes shape space and determine if differences between specimens (e.g., based on Procrustes distances) are consistent when using Sphaeroptica versus microCT models.

2. It is essential that error tests (between methods and/or observers) are also conducted using the 3D coordinates themselves. Currently, the error tests rely on linear distances between landmarks, which can be flawed. Different pairs of landmarks can have similar distances, especially in circular or elliptical structures like an insect's body. Therefore, the landmark positions may differ significantly without greatly affecting the linear measurements. This potential bias can be addressed by running error tests directly on the coordinate data. For instance, the authors could:

- A. First, calculate the percentage of mean deviation per landmark coordinate across repetitions within the same technology/tool (Sphaeroptica, microCT) and then compare the % error between technologies

- B. Then, also re-run the calculation% after Procrustes-superimposing the landmarks (together with those from the microCT models) to see if Procrustes distances between techniques (in%) are greater than within when they are all superimposed together.

3. The authors should also at least discuss whether Sphaeroptica could potentially reliably digitize sliding semi-landmarks along curves or surface patches, which are increasingly used in some fields and can provide more detailed information on 3D morphological variation.

4. The manuscript should include more specific information on the expected training time required to learn the process, whether prior knowledge of certain techniques (photogrammetry) is required, and whether observers need to be trained by someone familiar with the approach.

5. A major limitation of this new tool is its time-consuming nature compared to other techniques, related to the large amount of data generated per specimen. The authors candidly address this issue in Table 10 and Section 3.3, comparing across technologies/techniques, and suggest potential improvements. Further speeding up the processing would further enhance the practical value of this tool for analyzing larger sample sizes. Considering the high number of images, would some sort of a picture-sampling algorithm help, perhaps (without affecting the reliability of measurement)?

- Also, I would encourage the authors to briefly discuss the applicability on larger sample sizes.

6. The authors mention the limitation of movement artifacts that prevent landmark-based analysis of extremities (e.g., legs). While this is addressed in the Discussion with potential plans for improvement, such as using a robotic arm, other possible solutions should be briefly discussed (e.g., perhaps after ditigizing landmarks on extremities across pictures, or by filtering out images with considerable movement).

7. To my understanding, each landmark can be digitized on a single 2D image. Two images = 2 landmarks to use for measuring distance. If that is the case, it would be interesting and important to have some estimate of the variability (e.g. deviation %) of the same landmark position across different consequent images (where the same structure is visible). If such error occurs between images for the same landmark (which might theoretically be possible since each 2D image itself is lacking 3D information within it), this might mean that the choice of an exact image for the landmark may decrease measuring repeatability (e.g., if Observer 1 chooses image 4500 for digitizing landmark X, but Observer 2 chooses image 4503 for the same landmark). If the error is not considerable, then it would suggest that the choice of image does not affect the result. If the authors have saved this information (which image was used) for their already performed tests, error (% deviation in landmark coordinates) should be quite fast to calculate.

Addressing these concerns will strengthen the manuscript and enhance the utility of Sphaeroptica.

Reviewer #2: In their paper, the authors present their software “Sphaeroptica,” which facilitates 3D measurements from a sphere of 2D images, rather than from a 3D model calculated from these images. Although this might initially seem counter-intuitive, it is actually a very clever approach. This method allows for the inclusion of tiny details, such as small bristles or hairs that are visible in the photographs but often fail to be represented in a photogrammetry-based mesh.

Sphaeroptica is open source, with all resources and detailed manuals readily available. It works in tandem with popular photogrammetry systems like scAnt and DISC3D, enabling users to easily integrate the software into their existing workflows.

The authors demonstrate the effectiveness of Sphaeroptica by comparing its results with landmarking based on photogrammetry, structured light, and X-ray micro-CT. This section is detailed and documents the thorough evaluation process. The selection of test samples effectively reflects various challenges.

The paper is well-written, technically sound, and supported by convincing figures and tables that document the software's functionality and the evaluation results. I am convinced that Sphaeroptica is a very useful tool for anyone working on arthropod 3D geometric morphometrics. I congratulate the authors on their work and look forward to seeing this paper published.

Minor remarks:

Line 288: “Coleoptera” (capital C)

Line 296: “Coleoptera” (capital C); “Ovalisia” (italics); “sp.” (no italics)

Line 318: “and” (no italics)

Line 685: Consider using a different abbreviation than “CT,” as this usually stands for “computed tomography” and may cause confusion.

6. PLOS authors have the option to publish the peer review history of their article (what does this mean?). If published, this will include your full peer review and any attached files.

Reviewer #1: **Yes: **PD Dr. Fotios Alexandros Karakostis

Reviewer #2: No

---

## [Author Response · Author response to Decision Letter 0]

12 Sep 2024

Dear Editor, dear reviewers, 

Thank you for giving us the opportunity to submit a revised version of the manuscript “Sphaeroptica: A Tool for pseudo-3D Visualization and 3D Measurements on Arthropods” for publication. We appreciate the time and effort that you and the reviewers dedicated to providing feedback on our manuscript and are grateful for the insightful comments and valuable improvements to our paper. We have carefully considered the reviewers’ comments and made the necessary revisions to the manuscript.

Please see below, in red, for a point-by-point response to the reviewers’ comments and concerns.

Reviewer 1:

 1. Although the manuscript emphasizes 3D geometric morphometrics (GM), none of the analyses directly relate to GM. Landmark coordinates are used for linear measurements, but the coordinates were not extracted and superimposed to obtain shape variables. This process could be used to compare different specimens in Procrustes shape space and determine if differences between specimens (e.g., based on Procrustes distances) are consistent when using Sphaeroptica versus microCT models.

Author response: Indeed, the reviewer is correct. We did not apply 3D geometric morphometrics (GM) in this study, and therefore it should not be emphasized. We have revised the title and content to focus on the visualization tool and the linear measurements. However, we have also included a discussion on the potential applications of Sphaeroptica for GM, explaining how it could be used in future analyses.

2. It is essential that error tests (between methods and/or observers) are also conducted using the 3D coordinates themselves. Currently, the error tests rely on linear distances between landmarks, which can be flawed. Different pairs of landmarks can have similar distances, especially in circular or elliptical structures like an insect's body. Therefore, the landmark positions may differ significantly without greatly affecting the linear measurements. This potential bias can be addressed by running error tests directly on the coordinate data. For instance, the authors could:

 - A. First, calculate the percentage of mean deviation per landmark coordinate across repetitions within the same technology/tool (Sphaeroptica, microCT) and then compare the % error between technologies

 - B. Then, also re-run the calculation% after Procrustes-superimposing the landmarks (together with those from the microCT models) to see if Procrustes distances between techniques (in%) are greater than within when they are all superimposed together.

Author response: We appreciate the reviewer's insightful comment regarding the importance of error testing directly on 3D coordinates. While we acknowledge the potential benefits of this approach, our primary focus in this paper was to demonstrate the feasibility and accuracy of conducting 3D linear measurements of arthropod structures using Sphaeroptica. We believe that by concentrating on linear measurements, we effectively showcase the tool's potential for precise morphological analysis.

In Table 11, we have evaluated the repeatability of the linear measurements over 20 measurements for each technique, using standard deviation, percentage of deviation to average and the alpha value of repeatability test of Cronbach.

Additionally, for each specimen, we have added a table representing the absolute average difference and cumulated differences for each technique over all measurements of the specimens.

We recognize the limitations of relying solely on linear distances and the potential for inaccuracies in certain cases. As such, we plan to incorporate the suggested error tests based on 3D coordinates in future studies to further refine the capabilities of Sphaeroptica.

3. The authors should also at least discuss whether Sphaeroptica could potentially reliably digitize sliding semi-landmarks along curves or surface patches, which are increasingly used in some fields and can provide more detailed information on 3D morphological variation.

Author response: Sphaeroptica cannot measure semi-landmarks at the moment. This is part of future development. We have specified this in the manuscript.

 4. The manuscript should include more specific information on the expected training time required to learn the process, whether prior knowledge of certain techniques (photogrammetry) is required, and whether observers need to be trained by someone familiar with the approach.

Author response: We have added this to the manuscript.

5. A major limitation of this new tool is its time-consuming nature compared to other techniques, related to the large amount of data generated per specimen. The authors candidly address this issue in Table 10 and Section 3.3, comparing across technologies/techniques, and suggest potential improvements. Further speeding up the processing would further enhance the practical value of this tool for analyzing larger sample sizes. Considering the high number of images, would some sort of a picture-sampling algorithm help, perhaps (without affecting the reliability of measurement)?

 - Also, I would encourage the authors to briefly discuss the applicability on larger sample sizes. 

Author response: We appreciate the reviewer's concerns about processing time and agree with him that it is a limitation we need to overcome in future developments. As detailed in Section 3.3, we have compared Sphaeroptica's performance to other techniques and identified areas for optimization. Nevertheless, maintaining high image quality is crucial for optimal visualisation purposes and we believe it is one of the key advantages of the proposed method. However, we are actively exploring alternative methods to expedite the process, such as speeding up the stacking process using other processing tools, as well as considering other photogrammetric softwares.

While Part 4 emphasised Sphaeroptica's potential for various specimen sizes, a comprehensive analysis of larger non-arthropod specimens presents distinct challenges that warrant dedicated investigation. To maintain the manuscript's focus, we will address this topic in a separate work.

 6. The authors mention the limitation of movement artifacts that prevent landmark-based analysis of extremities (e.g., legs). While this is addressed in the Discussion with potential plans for improvement, such as using a robotic arm, other possible solutions should be briefly discussed (e.g., perhaps after ditigizing landmarks on extremities across pictures, or by filtering out images with considerable movement).

Author response: We understand the reviewer's concerns about movement artifacts affecting landmark-based analysis of extremities. As mentioned, we opted to avoid such analyses in this comparison to ensure data consistency across techniques. While movement artifacts can occur during image acquisition, with systems like ScAnt where the specimen moves, it is important to note that Sphaeroptica is a versatile tool capable of processing images from various acquisition methods, including robotic arms.

Our co-authors from HEIG-VD have developed a script to filter out images with excessive movement, which we hope to integrate into future versions of Sphaeroptica. Additionally, we will be exploring the potential of using landmark data from multiple images to mitigate the effects of movement on extremity analysis.

 7. To my understanding, each landmark can be digitized on a single 2D image. Two images = 2 landmarks to use for measuring distance. If that is the case, it would be interesting and important to have some estimate of the variability (e.g. deviation %) of the same landmark position across different consequent images (where the same structure is visible). If such error occurs between images for the same landmark (which might theoretically be possible since each 2D image itself is lacking 3D information within it), this might mean that the choice of an exact image for the landmark may decrease measuring repeatability (e.g., if Observer 1 chooses image 4500 for digitizing landmark X, but Observer 2 chooses image 4503 for the same landmark). If the error is not considerable, then it would suggest that the choice of image does not affect the result. If the authors have saved this information (which image was used) for their already performed tests, error (% deviation in landmark coordinates) should be quite fast to calculate.

Author response: The reviewer raised valid concerns about landmarks accuracy, but we fear there was some misunderstanding regarding our method. We have clarified that each landmark is indeed placed on at least two images. While acknowledging the potential influence of specific image choice on landmark precision, we believe that the use of multiple images by our operators, who selected images based on ease of landmark identification, minimises this effect.

Reviewer 2:

In their paper, the authors present their software “Sphaeroptica,” which facilitates 3D measurements from a sphere of 2D images, rather than from a 3D model calculated from these images. Although this might initially seem counter-intuitive, it is actually a very clever approach. This method allows for the inclusion of tiny details, such as small bristles or hairs that are visible in the photographs but often fail to be represented in a photogrammetry-based mesh.

Sphaeroptica is open source, with all resources and detailed manuals readily available. It works in tandem with popular photogrammetry systems like scAnt and DISC3D, enabling users to easily integrate the software into their existing workflows.

The authors demonstrate the effectiveness of Sphaeroptica by comparing its results with landmarking based on photogrammetry, structured light, and X-ray micro-CT. This section is detailed and documents the thorough evaluation process. The selection of test samples effectively reflects various challenges.

The paper is well-written, technically sound, and supported by convincing figures and tables that document the software's functionality and the evaluation results. I am convinced that Sphaeroptica is a very useful tool for anyone working on arthropod 3D geometric morphometrics. I congratulate the authors on their work and look forward to seeing this paper published.

Author response: Thank you!

Minor remarks:

Line 288: “Coleoptera” (capital C)

Line 296: “Coleoptera” (capital C); “Ovalisia” (italics); “sp.” (no italics)

Line 318: “and” (no italics)

Author response: Thank you to the reviewer for correcting these minor errors. All have been corrected within the paper.

Line 685: Consider using a different abbreviation than “CT,” as this usually stands for “computed tomography” and may cause confusion.

Author response: We agree with the reviewer that this abbreviation can be confusing. As a consequence, we switched from “CT” to “PT”, which stands for “processing time”. We hope this adds clarity to the text.

---

## [Decision Letter · Decision Letter 1]

27 Sep 2024

Sphaeroptica: A tool for pseudo-3D visualization and 3D measurements on arthropods

PONE-D-24-13247R1

Dear Dr. Mathys,

We’re pleased to inform you that your manuscript has been judged scientifically suitable for publication and will be formally accepted for publication once it meets all outstanding technical requirements.

Kind regards,

Giulia Pascoletti, Ph.D.

Academic Editor

PLOS ONE

Additional Editor Comments (optional):

Reviewers' comments:

Reviewer's Responses to Questions

**Comments to the Author**

1. If the authors have adequately addressed your comments raised in a previous round of review and you feel that this manuscript is now acceptable for publication, you may indicate that here to bypass the “Comments to the Author” section, enter your conflict of interest statement in the “Confidential to Editor” section, and submit your "Accept" recommendation.

Reviewer #1: All comments have been addressed

Reviewer #2: All comments have been addressed

2. Is the manuscript technically sound, and do the data support the conclusions?

Reviewer #1: Yes

Reviewer #2: Yes

3. Has the statistical analysis been performed appropriately and rigorously? 

Reviewer #1: Yes

Reviewer #2: N/A

4. Have the authors made all data underlying the findings in their manuscript fully available?

Reviewer #1: Yes

Reviewer #2: Yes

5. Is the manuscript presented in an intelligible fashion and written in standard English?

Reviewer #1: Yes

Reviewer #2: Yes

6. Review Comments to the Author

Reviewer #1: I would like to thank the authors for adequately addressing and/or discussing my concerns in their revised manuscript.

Reviewer #2: I thank the authors for revising their manuscript. The edits have improved the article considerably and, in my opinion, removed all ambiguities. It is very suitable for publication in PLOS ONE.

7. PLOS authors have the option to publish the peer review history of their article (what does this mean?). If published, this will include your full peer review and any attached files.

Reviewer #1: No

Reviewer #2: No

---

## [Editor Report · Acceptance letter]

10 Oct 2024

PONE-D-24-13247R1 

PLOS ONE

Dear Dr. Mathys, 

I'm pleased to inform you that your manuscript has been deemed suitable for publication in PLOS ONE. Congratulations! Your manuscript is now being handed over to our production team.

Kind regards, 

on behalf of

Dr. Giulia Pascoletti 

Academic Editor

PLOS ONE